# Adaptive Discretization for Consistency Models

**Jiayu Bai[1], Zhanbo Feng[2], Zhijie Deng[2], Tianqi Hou[3], Robert C. Qiu[1], Zenan Ling[1]***

[1]School of EIC, Huazhong University of Science and Technology
[2]School of Computer Science, Shanghai Jiao Tong University [3]Huawei

## Abstract

Consistency Models (CMs) have shown promise for efficient one-step generation. However, most existing CMs rely on manually designed discretization schemes, which can cause repeated adjustments for different noise schedules and datasets. To address this, we propose a unified framework for the automatic and adaptive discretization of CMs, formulating it as an optimization problem with respect to the discretization step. Concretely, during the consistency training process, we propose using local consistency as the optimization objective to ensure trainability by avoiding excessive discretization, and taking global consistency as a constraint to ensure stability by controlling the denoising error in the training target. We establish the trade-off between local and global consistency with a Lagrange multiplier. Building on this framework, we achieve adaptive discretization for CMs using the Gauss-Newton method. We refer to our approach as ADCMs. Experiments demonstrate that ADCMs significantly improve the training efficiency of CMs, achieving superior generative performance with minimal training overhead on both CIFAR-10 and ImageNet. Moreover, ADCMs exhibit strong adaptability to more advanced DM variants. Code is available at `https://github.com/rainstonee/ADCM`.

## 1  Introduction

Diffusion Models (DMs) [34, 9, 38, 18, 20] have achieved remarkable accomplishments in the field of data generation, including images [4, 29, 30], videos [10, 2, 41], audio [15, 28, 19] and 3D contents [40, 27, 22]. However, DMs require numerous iterations to achieve high-quality generation, significantly slowing sampling speed and making it resource-intensive. Recently, many fast-sampling methods for DMs have been proposed, including training-free methods [35, 14, 24, 48] and distillation-based approaches [25, 31, 46, 6, 26, 42, 45, 32]. However, these methods often sacrifice quality for faster sampling or incur substantial training overhead, which limits their practical application.

Consistency Models (CMs) [37, 7] offer significant advantages in addressing these challenges. CMs sample trajectories from the PF-ODE of DMs and aim to map each point on these trajectories to their corresponding endpoint. Through this approach, CMs achieve single-step generation while preserving the advantage of DMs, which improve generation quality by performing more iterations. CMs achieve the mapping to the endpoint by minimizing the distance between adjacent trajectory points. We refer to the selection of adjacent trajectory points as the *discretization* for CMs. Previous works [37, 36, 7, 23, 21] have shown that discretization is crucial for CMs' training. It determines the trainability and stability of CMs: poor trainability can impair final performance, while instability during training may slow convergence or even lead to divergence. A suboptimal discretization strategy may lead to an imbalance between trainability and stability [7]. It may also cause CMs to overly focus on training within a specific time interval, leading to a loss of consistency [17]. To mitigate these challenges and ensure a balanced training process, most existing works adopt empirical discretization strategies, which require manual adjustments based on different noise schedules and datasets [7].

---

*Corresponding Author: Zenan Ling (lingzenan@hust.edu.cn).

39th Conference on Neural Information Processing Systems (NeurIPS 2025).

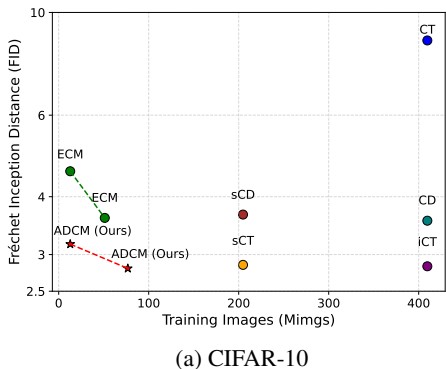
(a) CIFAR-10

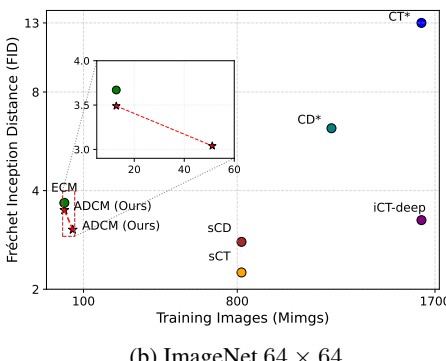
(b) ImageNet $64 \times 64$

Figure 1: ADCMs significantly improve the **training efficiency** on both (a) unconditional CIFAR-10 and (b) class-conditional ImageNet $64 \times 64$. ADCMs achieve superior generation quality with only a minimal amount of training data. $*$ indicates a smaller model.

Our fundamental goal is to adaptively determine the discretization strategy for CMs' training[2], considering both trainability and stability, thereby improving the training efficiency and final performance of CMs. First, we propose that the discretization step should minimize the optimization objective of CMs, i.e., *local consistency*, to ensure their trainability. Second, the discretization step controls the denoising error in the training target of CMs, which affects the *global consistency*. Excessive denoising error can lead to instability in CMs' training, thereby degrading the efficiency of CMs. Hence, we introduce global consistency as a constraint to ensure stability. To adaptively balance trainability and stability, we formulate local and global consistency as a constrained optimization problem and relax it via the Lagrangian method, introducing a Lagrange multiplier to express the trade-off between them. To achieve adaptive discretization, we propose using the Gauss-Newton method to obtain an analytical solution to the optimization problem. We refer to our method as Adaptive Discretization for Consistency Models (ADCMs). Our analysis reveals that ADCMs adaptively adjust discretization steps by jointly considering local and global consistency, thus achieving a balanced trade-off between trainability and stability.

In our experiments, ADCMs significantly improve the training efficiency and final performance of CMs. On unconditional CIFAR-10, as shown in Figure 1a, ADCMs exhibit outstanding training efficiency. We achieve a 1-step FID of 3.16 with only a training budget of 12.8M images. In contrast, ECM [7], the most efficient CM from previous work, requires 51.2M images to reach a FID of 3.60. Moreover, we attain a 1-step FID of 2.80 using only 76.8M images, outperforming iCT [36], which requires 409.6M images to achieve comparable performance. On class-conditional ImageNet $64 \times 64$, as shown in Figure 1b, ADCMs significantly reduce the training overhead under the same model size. ADCMs achieve a 1-step FID of 3.49 with a training budget of only 12.8M images. When the training budget increases to 51.2M images, ADCMs achieve a competitive 1-step FID of 3.04.

**Contributions.**  Our contributions are summarized as follows.

- We provide a unified framework for the discretization for CMs. Starting from local consistency and global consistency, we investigate the impact of discretization steps on the trainability and stability of CMs. Guided by these two principles, we formulate a constrained optimization problem for selecting the discretization step. Previous discretization methods can be regarded as special cases of our framework.

- Based on this framework, we propose Adaptive Discretization for Consistency Models (ADCMs). First, we relax the optimization problem using the Lagrangian method and establish a trade-off between the trainability and stability of CMs through the Lagrange multiplier. Then, we employ the Gauss-Newton method to derive an analytical solution to the optimization problem, enabling adaptive discretization steps that effectively balance

---

[2]In particular, we focus on Consistency Training (CT) [37] over Consistency Distillation (CD) due to its superior empirical performance and its ability to bypass numerical solvers by directly leveraging training data for unbiased score estimation.

local and global consistency. Additionally, we introduce an adaptive loss function to further improve performance.

- Our experiments demonstrate that ADCMs significantly improve the training efficiency, while achieving competitive performance in one-step generation. On CIFAR-10, ADCMs achieve superior results using less than $25\%$ of the training budget compared to previous works. On ImageNet, ADCMs also demonstrate strong performance with minimal training overhead. Furthermore, ADCMs adapt to advanced variants of DMs such as Flow Matching without manual adjustments.

## 1.1 Related Works

**Consistency Models.** Consistency Models were first proposed by [37], achieving the distillation of Diffusion Models by mapping any point on the PF-ODE trajectory to the endpoint of the trajectory. To accomplish this, it proposed sampling adjacent points on the trajectory and enforcing that the output near the noise end approximates the output near the data end. It divided CMs into two categories: Consistency Distillation (CD) and Consistency Training (CT), corresponding to sampling trajectory points using a pretrained DM and the forward diffusion process, respectively. iCT [36] explored the potential of CT, as it does not require a pretrained DM to sample trajectory points, thus supporting training from scratch. ECM [7] discovered that initializing the CM with a pretrained DM can effectively accelerate its training speed. TCM [17] discovered that CMs struggle to map the entire trajectory using a single model. Therefore, it proposed a two-stage approach for CMs, enabling CMs to focus on learning tasks from different time intervals separately. CTM [13] and Shortcut Models [5] aimed to make CMs capable of mapping to any point on the trajectory, not just the endpoint, by introducing an additional time condition to assist the model's learning.

**Discretization for CMs.** The training of CMs fundamentally relies on the selection of adjacent trajectory points, a process we refer to as the discretization for CMs. Various discretization strategies have been explored in previous works. iCT [36] proposed segmenting time based on the sampling steps of DMs [11]. ECM [7] introduced a decoupled approach, employing two functions: one to determine the overall magnitude of discretization steps and another to regulate their distribution over time. Both iCT and ECM adopt exponentially decreasing time steps to enhance the stability of CMs' training. Alternatively, CCM [21] introduced an adaptive discretization scheme by iteratively solving for the discretization step based on a Peak Signal-to-Noise Ratio (PSNR) threshold, ensuring a more balanced training across different times. sCM [23] formulated an "infinite" discretization approach, where adjacent trajectory points become infinitesimally close, transforming their distance into the first-order time derivative. However, sCM observed that this discretization scheme suffers from stability issues and proposed modifications to both the noise schedule and the neural network architecture, among other refinements, to ensure stable training.

## 2 Preliminaries

### 2.1 Diffusion Models

Given a dataset with an underlying distribution $p_{\text{data}}$, DMs generate samples by learning to reverse a noising process that progressively adds random Gaussian noise to the data, eventually transforming it into pure noise. Specifically, for a data sample $\boldsymbol{x}_0 \sim p_{\text{data}}$ and a noise sample $\boldsymbol{z} \sim \mathcal{N}(0, \boldsymbol{I})$, the diffusion process is defined as:

$$\boldsymbol{x}_t = \alpha_t \boldsymbol{x}_0 + \beta_t \boldsymbol{z}$$

where $t \in [\epsilon, T]$, and $\epsilon$ is a small value used to prevent numerical errors. [38] proposes that the diffusion process can be modeled as a forward SDE, which is then denoised step-by-step using the corresponding probability flow ODE (PF-ODE). DMs utilize a time-dependent neural network (NN) to predict the unknown $\boldsymbol{x}_0$ in the PF-ODE. The optimization objective of DMs is given by:

$$\min_{\theta} \mathbb{E}_{\boldsymbol{x}_0, \boldsymbol{z}, t}[w(t) \cdot \|f_\theta(\boldsymbol{x}_t) - \boldsymbol{x}_0\|_2^2] \tag{1}$$

where $f_\theta(\boldsymbol{x}_t) = c_{\text{skip}}(t)\boldsymbol{x}_t + c_{\text{out}}(t)F_\theta\big(c_{\text{in}}(t)\boldsymbol{x}_t, c_{\text{noise}}(t)\big)$, $w(t)$ is a weighting function and $F_\theta$ is an NN with parameters $\theta$. We write $f(\boldsymbol{x}_t, t)$ as $f(\boldsymbol{x}_t)$ for simplicity. Most DMs, including DDPM [9], EDM [11], and Flow Matching [18], have training objectives that can be equivalently expressed as Eq. (1) through the design of precondition $c_{\text{skip}}(t)$ and $c_{\text{out}}(t)$.

## 2.2 Consistency Models

CMs [37, 23] aim to map any point $\boldsymbol{x}_t$ on the PF-ODE trajectory to the corresponding data $\boldsymbol{x}_0$, i.e., $f_\theta(\boldsymbol{x}_t, t) = \boldsymbol{x}_0$. To achieve this, (1) at $t = 0$, CMs require that $f_\theta$ satisfy the boundary condition $f_\theta(\boldsymbol{x}_\epsilon, \epsilon) \equiv \boldsymbol{x}_0$, which implies that $c_{\text{skip}}(\epsilon) = 1$ and $c_{\text{out}}(\epsilon) = 0$; (2) for $t > 0$, CMs are trained to produce consistent outputs for any two adjacent points on the PF-ODE trajectory. Specifically, the optimization objective of CMs is given by:

$$\min_\theta \mathbb{E}_{\boldsymbol{x}_0, \boldsymbol{z}, t}[w(t) \cdot \|f_\theta(\boldsymbol{x}_t) - f_{\theta^-}(\boldsymbol{x}_{t-\Delta t})\|_2^2] \tag{2}$$

where $\theta^-$ stands for $\text{stopgrad}(\theta)$ and $\Delta t$ is the time interval that defines the adjacent time step corresponding to a given time $t$, which in turn determines the training target on the PF-ODE trajectory. When retrieving the adjacent point $\boldsymbol{x}_{t-\Delta t}$, we adopt Consistency Training (CT) paradigm [37] which enables unbiased estimation of the score function, expressed as $\frac{\boldsymbol{x}_t - \alpha_t \boldsymbol{x}_0}{\beta_t^2}$. This approach eliminates the numerical errors introduced by solvers required for Consistency Distillation (CD).

The choice of $\Delta t$, referred to as the *discretization* of CMs, plays a crucial role in their training [37, 21, 23]. Previous works have proposed various discretization strategies, which can be categorized into two types, as outlined below.

**Discrete-CMs.** When $\Delta t > 0$ is not infinitesimally small, CMs fall to discrete-CMs. Discrete-CMs require careful planning of the discretization schedule. iCT [36] divides the time interval $[\epsilon, T]$ into multiple segments using the sampling time steps in DMs [11], i.e., $\mathbb{T} = \{t_0, \ldots, t_N\}$ where $t_0 = \epsilon$ and $t_N = T$, and samples adjacent time points within $\mathbb{T}$ as $t$ and $t - \Delta t$. iCT proposes that the discretization of CMs requires meticulous planning and designs time steps that decrease progressively during training. ECM [7] also adopts dynamic time step scheduling. Unlike iCT, ECM samples $t$ in continuous time and then maps the corresponding time step through a manually designed function. CCM [21] points out that the discretization step size affects the training difficulty of CMs at different times. CCM proposes setting a PSNR threshold for CMs and solving the PF-ODE iteratively until the selected time steps satisfy this threshold.

**Continuous-CMs.** Taking the limit $\Delta t \to 0$, CMs fall to continuous-CMs, which can be considered equivalent to "infinite" discretization. [37] proves that continuous-CMs can be optimized with:

$$\nabla_\theta \mathbb{E}_{\boldsymbol{x}_0, \boldsymbol{z}, t}\left[w(t) f_\theta^\top(\boldsymbol{x}_t) \frac{\mathrm{d}f_{\theta^-}(\boldsymbol{x}_t)}{\mathrm{d}t}\right] \tag{3}$$

which is a continuous version of Eq. (2). Continuous-CMs effectively avoid the discretization schedule of discrete-CMs. However, continuous-CMs often face significant instability challenges. sCM [23] addresses this instability for a specific DM with a specialized noise schedule, but it remains unclear that how to address the instability for continuous CMs under a more general setting.

Overall, the choice of discretization step $\Delta t$ is still challenging. If $\Delta t$ is too large, CMs struggle to learn meaningful information, while if it is too small, instability issues arise [23]. Additionally, how $\Delta t$ varies *w.r.t.* $t$ determines the training emphasis at different times [36, 7], and suboptimal strategies can lead the model to focus on specific time intervals, negatively impacting overall performance. Although various discretization strategies have been proposed, they often fail to identify the optimal $\Delta t$ for each time step. On the one hand, existing discrete-CMs lack adaptive adjustment capabilities, requiring additional modeling and hyperparameter tuning for different noise schedules and datasets. On the other hand, continuous-CMs avoid discrete time steps by treating all time steps equally, but not all are equally important for effective training [36, 17]. This limits training efficiency.

## 3 Methodology

### 3.1 ADCMs: Adaptive Discretization for Consistency Models

In Section 2.2, we illustrate the importance of discretization in training CMs. In this study, our fundamental goal is to determine which discretization strategy is most beneficial for CMs' training, i.e., the discretization step $\Delta t$ for a *given* time $t$. When we fix the NN's parameters $\theta^- = \text{stopgrad}(\theta)$ and time $t$, we aim to find an optimal $\Delta t$ that contributes the most to the following training objective of CMs:

$$\mathbb{E}_{\boldsymbol{x}_0, \boldsymbol{z}}[\|f_{\theta^-}(\boldsymbol{x}_t) - f_{\theta^-}(\boldsymbol{x}_{t-\Delta t})\|_2^2]. \tag{4}$$

We define Eq. (4) as *local consistency* as it reflects the properties of CMs in a local interval. First, we need that the objective in Eq. (4) is trainable. To achieve this, we need to choose an appropriate $\Delta t$ such that the objective is as small as possible, thereby satisfying local consistency, namely:

$$\min_{\Delta t} \mathbb{E}_{\boldsymbol{x}_0, \boldsymbol{z}} \left[ \| f_{\theta^-}(\boldsymbol{x}_t) - f_{\theta^-}(\boldsymbol{x}_{t-\Delta t}) \|_2^2 \right]. \tag{5}$$

It can be observed that when $\Delta t = 0$, the local consistency in Eq. (5) is minimized. This implies that we need to choose $\Delta t$ as small as possible. However, previous works [37, 7, 23] have shown that when $\Delta t$ is too small, CMs face severe stability issues, which slows down convergence and may even lead to divergence. The underlying reason is that the practical training target, i.e., $f_{\theta^-}(\boldsymbol{x}_{t-\Delta t})$, fails to precisely denoise $\boldsymbol{x}_{t-\Delta t}$ to the ground-truth $\boldsymbol{x}_0$, leading to the global denoising error quantified as:

$$\mathbb{E}_{\boldsymbol{x}_0, \boldsymbol{z}} \left[ \| f_{\theta^-}(\boldsymbol{x}_{t-\Delta t}) - \boldsymbol{x}_0 \|_2^2 \right]. \tag{6}$$

*Remark* 3.1. This denoising error is also an upper bound on the squared Wasserstein-2 distance between $p_{\text{data}}$ and the data distribution generated by $f_{\theta^-}$ at time $t - \Delta t$. Moreover, this error can be regarded as a lower bound on the accumulated error from previous time steps, namely:

$$\sqrt{\mathbb{E}_{\boldsymbol{x}_0, \boldsymbol{z}} \left[ \| f_{\theta^-}(\boldsymbol{x}_{t-\Delta t}) - \boldsymbol{x}_0 \|_2^2 \right]} \leq \sum_{i=1}^{k} \sqrt{\mathbb{E}_{\boldsymbol{x}_0, \boldsymbol{z}} \left[ \| f_{\theta^-}(\boldsymbol{x}_{t_i}) - f_{\theta^-}(\boldsymbol{x}_{t_i - \Delta t_i}) \|_2^2 \right]},$$

where $\Delta t_i$ is the discretization step corresponding to time $t_i$, satisfying $t_i - \Delta t_i = t_{i-1}$.

We define Eq. (6) as *global consistency* because it reflects the global properties of CMs. Excessive denoising error will cause CMs to optimize in the wrong direction, which leads to instability in CMs' training. Therefore, we propose that when selecting the discretization step $\Delta t$, it is necessary to ensure that the denoising error is constrained, namely:

$$\text{Find} \quad \Delta t, \quad \text{s.t.} \quad \mathbb{E}_{\boldsymbol{x}_0, \boldsymbol{z}} \left[ \| f_{\theta^-}(\boldsymbol{x}_{t-\Delta t}) - \boldsymbol{x}_0 \|_2^2 \right] \leq \delta \tag{7}$$

where $\delta$ is an upper bound that needs to be set manually. Clearly, when $\Delta t$ takes the maximum value $t - \epsilon$, due to the boundary condition $f_\theta(\boldsymbol{x}_\epsilon, \epsilon) \equiv \boldsymbol{x}_0$, the constraint in Eq. (7) will be satisfied regardless of the value of $\delta$. This implies that we need to choose the largest possible $\Delta t$.

Notably, the optimization objective in Eq. (5) and the constraint in Eq. (7) respectively impose opposite guidance for $\Delta t$. When $\Delta t$ is extremely small, the local consistency error in Eq. (5) is minimized, making it easy for CMs to optimize. However, this may cause the constraint in Eq. (7) to exceed its upper bound. Conversely, when $\Delta t$ is extremely large, the constraint in Eq. (7) will be easily satisfied, but it may cause the optimization objective in Eq. (5) to become too large and difficult to optimize. Therefore, we propose a constrained optimization objective to achieve a trade-off between Eq. (5) and Eq. (7), which is given by:

$$\min_{\Delta t} \quad \mathbb{E}_{\boldsymbol{x}_0, \boldsymbol{z}} \left[ \| f_{\theta^-}(\boldsymbol{x}_t) - f_{\theta^-}(\boldsymbol{x}_{t-\Delta t}) \|_2^2 \right], \quad \text{s.t.} \quad \mathbb{E}_{\boldsymbol{x}_0, \boldsymbol{z}} \left[ \| f_{\theta^-}(\boldsymbol{x}_{t-\Delta t}) - \boldsymbol{x}_0 \|_2^2 \right] \leq \delta. \tag{8}$$

We denote the optimization objective $\mathbb{E}_{\boldsymbol{x}_0, \boldsymbol{z}}[\| f_{\theta^-}(\boldsymbol{x}_t) - f_{\theta^-}(\boldsymbol{x}_{t-\Delta t}) \|_2^2]$ as $\mathcal{L}_{\text{local}}$, as it focuses on the local consistency information of CMs and controls the local consistency error for CMs. Therefore, minimizing $\mathcal{L}_{\text{local}}$ effectively improves the effectiveness of CMs. We denote the constraint $\mathbb{E}_{\boldsymbol{x}_0, \boldsymbol{z}}[\| f_{\theta^-}(\boldsymbol{x}_{t-\Delta t}) - \boldsymbol{x}_0 \|_2^2]$ as $\mathcal{L}_{\text{global}}$ as it focuses on the global consistency and controls the denoising error in the training objective. Consequently, $\mathcal{L}_{\text{global}}$ helps CMs effectively eliminate denoising error, find accurate optimization targets, and thus improve training stability and efficiency. Our goal is to ensure both the global consistency and the local consistency simultaneously, enabling an adaptive adjustment of CMs' discretization. However, directly optimizing the constrained optimization problem in Eq. (8) is challenging. To address this, we apply the Lagrange multiplier method to relax the problem, yielding the following formulation:

$$\Delta t^* = \arg\min_{\Delta t} \mathbb{E}_{\boldsymbol{x}_0, \boldsymbol{z}}[\mathcal{L}_{\text{local}}(t, \Delta t) + \lambda \mathcal{L}_{\text{global}}(t, \Delta t)]. \tag{9}$$

Here, the Lagrange multiplier $\lambda$ acts as a weighting factor balancing the local consistency and the global consistency of CMs. We aim for $\lambda$ to be a constant independent of $t$, ensuring that the focus on trainability and stability remains consistent across different time scales. Typically, we set $\lambda \ll 1$, as ensuring whether CMs are trainable is of greater importance compared to their stability. We refer to our approach as Adaptive Discretization for Consistency Models (ADCMs), as shown in Figure 2. We find that previous discretization strategies can be unified in ADCMs. We summarize this as follows.

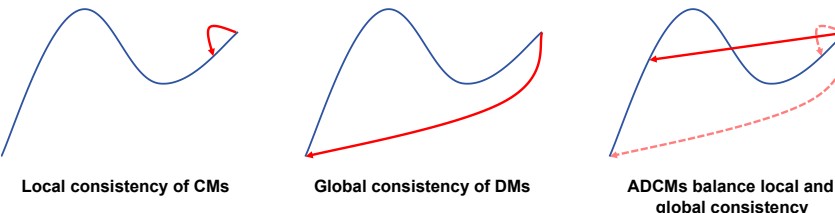

**Local consistency of CMs**     **Global consistency of DMs**     **ADCMs balance local and global consistency**

Figure 2: Discretization strategies of different models. CMs consider only local consistency during discretization, while DMs consider only global consistency. ADCMs balance local and global consistency and adaptively adjust the discretization step size based on the information from both.

*Remark* 3.2. DMs, discrete-CMs and continuous-CMs can be viewed as special cases of Eq. (9). Specifically,

- DMs [11, 18]: Choose the maximum optimization step $\Delta t = t - \epsilon$. this corresponds to set $\lambda \to \infty$ in our framework.

- Discrete-CMs:

  - CM [37], iCT [36], ECM [7]: These methods consider the smoother trajectory changes near the noise end and empirically choose a discretization step size that monotonically increases *w.r.t.* $t$. This is equivalent to estimating Eq. (9) empirically in our framework.
  - CCM [21]: CCM ensures that for all $\boldsymbol{x}_0, \boldsymbol{z}, \mathcal{L}_e$ is always less than a certain threshold $\delta$. Since an analytical solution cannot be obtained directly, CCM requires iterative solving for all $\boldsymbol{x}_0, \boldsymbol{z}, t$. This is equivalent to making $\mathcal{L}_{\text{local}}$ a constant in our framework.

- Continuous-CMs [23]: Choose the minimum optimization step $\Delta t \to 0$. This is equivalent to set $\lambda = 0$ in our framework.

**Analytical Solution.** To achieve an adaptive solution for the discretization step, we propose using the Gauss-Newton method to directly derive an analytical solution to the optimization problem in Eq. (9). Since we assign a higher weight to local consistency in the objective, we approximate $f_{\theta^-}(\boldsymbol{x}_{t-\Delta t})$ using its first-order Taylor expansion, which is:

$$f_{\theta^-}(\boldsymbol{x}_{t-\Delta t}) \approx f_{\theta^-}(\boldsymbol{x}_t) - \boldsymbol{v}\Delta t, \quad \boldsymbol{v} = \nabla_{\boldsymbol{x}_t} f_{\theta^-} \cdot \frac{\mathrm{d}\boldsymbol{x}_t}{\mathrm{d}t} + \partial_t f_{\theta^-}$$

where $\boldsymbol{v}$ can be efficiently computed via the Jacobian-vector product (JVP) for $f_{\theta^-}(\cdot, \cdot)$, evaluated at input vector $(\boldsymbol{x}_t, t)$ and tangent vector $(\frac{\mathrm{d}\boldsymbol{x}_t}{\mathrm{d}t}, 1)$, following the method in [23]. Under this approximation, the optimization problem is transformed into a least-squares problem, whose optimal solution is given by:

$$\Delta t^* = \frac{\lambda}{1+\lambda} \frac{\mathbb{E}_{\boldsymbol{x}_0, \boldsymbol{z}}[\boldsymbol{v}^\top (f_{\theta^-}(\boldsymbol{x}_t) - \boldsymbol{x}_0)]}{\mathbb{E}_{\boldsymbol{x}_0, \boldsymbol{z}}[\boldsymbol{v}^\top \boldsymbol{v}]}. \tag{10}$$

From the expression of the discretization step, we have the following observations:

1. The optimal discretization step is inversely proportional to the magnitude of the Jacobian. This indicates that the output of the current network may vary significantly, and $\mathcal{L}_{\text{local}}$ could be very large. Therefore, a smaller step size is required to ensure effectiveness.

2. The optimal discretization step is proportional to $\|f_{\theta^-}(\boldsymbol{x}_t) - \boldsymbol{x}_0\|_2$, which is an effective estimate of $\mathcal{L}_{\text{global}}$. This indicates that the denoising error may be very large at this time, and therefore, a larger step size is required to ensure stability.

3. The optimal discretization step is proportional to the linear correlation between $\boldsymbol{v}$ and $f_{\theta^-}(\boldsymbol{x}_t) - \boldsymbol{x}_0$. This indicates that when $\boldsymbol{v}$ and $f_{\theta^-}(\boldsymbol{x}_t) - \boldsymbol{x}_0$ tend toward linearity, the direction of local optimization aligns more closely with the direction of global optimization, allowing for the use of a larger step size.

---

**Algorithm 1** Adaptive Discretization for Consistency Models

---

**Input:** dataset $\mathcal{D}$, diffusion parameter $\alpha_t$ and $\beta_t$, time range $[\epsilon, T]$, network parameter $\theta$, weighting factor $\lambda$, update frequency $m$, batch size $b$
$\theta^- \leftarrow \theta$
**repeat**
   Initialize an empty set $\mathbb{T}$ and $t \leftarrow T$
   **repeat**
      Append $t$ to $\mathbb{T}$
      Sample mini-batch $\boldsymbol{x}_0 \sim \mathcal{D}$, $\boldsymbol{z} \sim \mathcal{N}(0, \boldsymbol{I})$
      $\boldsymbol{x}_t \leftarrow \alpha_t \boldsymbol{x}_0 + \beta_t \boldsymbol{z}$
      Calculate $\Delta t^*$ through Eq. (10)
      $t \leftarrow t - \Delta t^*$
   **until** $t \leq \epsilon$
   Append $\epsilon$ to $\mathbb{T}$
   **for** $k = 1$ **to** $m$ **do**
      Sample mini-batch $\boldsymbol{x}_0 \sim \mathcal{D}$, $\boldsymbol{z} \sim \mathcal{N}(0, \boldsymbol{I})$ and adjacent time points $t, t - \Delta t^* \sim \mathbb{T}$
      $\boldsymbol{x}_t \leftarrow \alpha_t \boldsymbol{x}_0 + \beta_t \boldsymbol{z}$, $\boldsymbol{x}_{t-\Delta t^*} \leftarrow \alpha_{t-\Delta t^*} \boldsymbol{x}_0 + \beta_{t-\Delta t^*} \boldsymbol{z}$
      Calculate loss $\mathcal{L}$ through Eq. (11)
      Update $\theta$ using $\mathcal{L}$
   **end for**
**until** Convergence

---

The above analysis demonstrates that the proposed discretization step can be adaptively adjusted based on the current state of the NN, considering both $\mathcal{L}_{\text{global}}$ and $\mathcal{L}_{\text{local}}$. As a result, we achieve an adaptive balance between the stability and trainability of CMs at different times through Eq. (10). Starting from $t = T$, we iteratively solve the optimization problem to derive the adaptively optimal time segmentation $\mathbb{T} = \{t_1^*, \ldots, t_N^*\}$.

### 3.2 Putting ADCMs into Practice: Further Training Strategies

**Adaptive Weighting Function.** Through the analysis in Section 3.1, we know that $\mathcal{L}_{\text{global}}$ fundamentally determines the training stability of CMs at the current time $t$. However, during the training of CMs, the NN only optimizes for $\mathcal{L}_{\text{local}}$. Therefore, to further balance the impact of $\mathcal{L}_{\text{global}}$ over time, we propose the following adaptive weighting function:

$$w(t) = \frac{1}{\mathcal{L}_{\text{global}}} = \frac{1}{\|f_{\theta^-}(\boldsymbol{x}_{t-\Delta t}) - \boldsymbol{x}_0\|_2^2}.$$

When $\mathcal{L}_{\text{global}}$ is very large, the training of CMs will suffer from instability. Therefore, a smaller weighting is needed. On the other hand, when $\mathcal{L}_{\text{global}}$ is small, the CMs' training objective aligns closely with the true target, and thus a larger weighting is required.

**Adaptive Distance Metric.** Previous works [7, 36] have shown that compared to the squared $L_2$ metric, Pseudo-Huber metric can effectively reduce training variance, which is given by:

$$d(\boldsymbol{x}, \boldsymbol{y}) = \sqrt{\|\boldsymbol{x} - \boldsymbol{y}\|_2^2 + c^2} - c$$

where $c$ is a constant. ADCMs also use Pseudo-Huber metric for training. At the same time, in order to ensure the consistency of the distance function, we have similarly modified the adaptive weighting function. The overall loss function of ADCMs can be expressed as:

$$\min_\theta \mathbb{E}_{\boldsymbol{x}_0, \boldsymbol{z}, t} \left[ \frac{\sqrt{\|f_\theta(\boldsymbol{x}_t) - f_{\theta^-}(\boldsymbol{x}_{t-\Delta t^*})\|_2^2 + c^2} - c}{\sqrt{\|f_{\theta^-}(\boldsymbol{x}_{t-\Delta t^*}) - \boldsymbol{x}_0\|_2^2 + c^2} - c} \right] \tag{11}$$

where $\Delta t^*$ is obtained with Eq. (10). See Appendix B for more discussion on loss function design.

**Putting It Together.** We alternately optimize the time segmentation $\mathbb{T}$ and the NN's parameters $\theta$ during training. We typically update $\mathbb{T}$ after updating $\theta$ for $m = 25000$ times, as the changes in the

Table 1: Sample quality on unconditional CIFAR-10 and class-conditional ImageNet $64 \times 64$. $*$ indicates additional training costs.

| Method | CIFAR-10 NFE ($\downarrow$) | CIFAR-10 FID ($\downarrow$) | ImageNet $64 \times 64$ NFE ($\downarrow$) | ImageNet $64 \times 64$ FID ($\downarrow$) |
|---|---|---|---|---|
| **Diffusion Models** | | | | |
| DDPM [9] | 1000 | 3.17 | 250 | 11.0 |
| EDM [11] | 35 | 1.97 | 511 | 1.36 |
| DPM-Solver [24] | 10 | 4.70 | 20 | 3.42 |
| 1-Rectified Flow [20] | 127 | 2.58 | – | – |
| ADM [4] | – | – | 250 | 2.07 |
| EDM2-S [12] | – | – | 63 | 1.58 |
| EDM2-XL [12] | – | – | 63 | 1.33 |
| **Joint Training** | | | | |
| StyleGAN-XL [33] | 1 | 1.52 | 1 | 1.52 |
| SiD [47] | 1 | 1.92 | 1 | 1.52 |
| CTM [13] | 1 | 1.87 | 1 | 1.92 |
| CCM [21] | 1 | 1.64 | 1 | 2.18 |
| Consistency-FM [43] | 2 | 1.69 | 2 | 1.62 |
| DMD2 [44] | – | – | 1 | 1.28 |
| **Diffusion Distillation** | | | | |
| DFNO (LPIPS) [46] | 1 | 3.78 | 1 | 7.83 |
| PID (LPIPS) [39] | 1 | 3.92 | 1 | 9.49 |
| TRACT [1] | 1 | 3.78 | 1 | 7.43 |
| PD [31] | 1 | 8.34 | 1 | 10.70 |
| 2-Rectified Flow [20] | 1 | 4.85 | – | – |
| **Consistency Models** | | | | |
| CD (LPIPS) [37] | 1 | 3.55* | 1 | 6.20* |
| CT [37] | 1 | 8.70 | 1 | 13.0 |
| iCT [36] | 1 | 2.83 | 1 | 4.02 |
| iCT-deep [36] | 1 | 2.51* | 1 | 3.25 |
| ECM [7] | 1 | 3.60 | 1 | 2.49* |
| TCM [17] | 1 | 2.46* | 1 | 2.20* |
| sCD [23] | 1 | 3.66 | 1 | 2.44* |
| sCT [23] | 1 | 2.85 | 1 | 2.04* |
| ADCM (ours) | 1 | 2.80 | 1 | 3.04 |

Table 2: Training efficiency on CIFAR-10.

| Unconditional CIFAR-10 | | |
|---|---|---|
| **Method** | **Training Budget (Mimgs)** | **1-Step FID ($\downarrow$)** |
| CD (LPIPS) | 409.6 | 3.55 |
| iCT | 409.6 | 2.83 |
| sCT (TrigFlow) | 204.8 | 2.85 |
| sCT (VE) | 51.2 | 4.18 |
| ECM | 12.8 | 4.54 |
| ECM | 51.2 | 3.60 |
| ADCM (Ours) | 12.8 | 3.16 |
| ADCM (Ours) | 76.8 | **2.80** |

Table 3: Training efficiency on ImageNet $64 \times 64$.

| Class-Conditional ImageNet $64 \times 64$ | | | |
|---|---|---|---|
| **Method** | **Model Size** | **Training Budget (Mimgs)** | **1-Step FID ($\downarrow$)** |
| CD (LPIPS) | $1\times$ | 1228.8 | 6.20 |
| iCT | $1\times$ | 1638.4 | 4.02 |
| iCT-deep | $2\times$ | 1638.4 | 3.25 |
| sCT (TrigFlow) | $2\times$ | 819.2 | 2.25 |
| ECM | $1\times$ | 12.8 | 5.51 |
| ECM | $2\times$ | 12.8 | 3.67 |
| ADCM (Ours) | $1\times$ | 12.8 | 5.12 |
| ADCM (Ours) | $1\times$ | 25.6 | 4.65 |
| ADCM (Ours) | $1\times$ | 51.2 | 4.23 |
| ADCM (Ours) | $2\times$ | 12.8 | 3.49 |
| ADCM (Ours) | $2\times$ | 25.6 | 3.28 |
| ADCM (Ours) | $2\times$ | 51.2 | 3.04 |

NN are relatively slow. Before training the network, we fix its parameters and perform simulation-based optimization starting from $t = T$. We iteratively update $t$ using Eq. (10) until $t = \epsilon$, recording the optimization process as $\mathbb{T} = \{t_1^*, \ldots, t_N^*\}$. We observe that during the optimization process, the expectation in Eq. (10) is well approximated using a single mini-batch. This is because we do not require precise step sizes, only the trend of their change over time $t$. Subsequently, we fix the time segmentation and optimize the NN. The detailed process is illustrated in Algorithm 1.

## 4 Experiments

To validate the effectiveness of ADCMs, we perform unconditional and class-conditional generation experiments on CIFAR-10 [16] and ImageNet $64 \times 64$ [3], respectively. For CIFAR-10, we initialize CMs with pretrained DM from [11]. For ImageNet $64 \times 64$, we adopt the pretrained DM from [12]. If not otherwise specified, our experiments are conducted under VE SDE [38] settings. We evaluate the sample quality using FID [8] and measure the generation speed using the number of function evaluations (NFEs).

We compare ADCMs with different generative models, as shown in Table 1. FIDs with $*$ indicate that they have additional training costs compared to other CMs, such as a larger model or an auxiliary model used during training. Experiments show that ADCMs achieve high-quality single-step generation without additional training costs. See Appendix C for multi-step generation results.

### 4.1 Efficiency of ADCMs

We evaluate the training efficiency of ADCMs on both unconditional CIFAR-10 and class-conditional ImageNet $64 \times 64$. For CIFAR-10, we use a unified model size and measure computational cost by the total number of training images. For ImageNet $64 \times 64$, both model size and training budgets are taken into consideration. TCM [17] is excluded from the comparison since its two-stage strategy introduces significant training overhead. For a fair comparison, we reproduce some baseline results, as detailed in Appendix A.3.

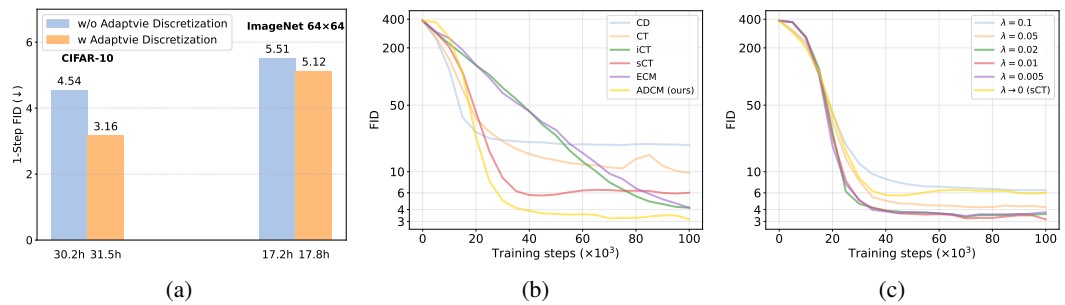

Figure 3: (a) Training time cost of ADCMs. (b) Training dynamics of different discretization methods. Compared to other CMs' approaches, ADCMs have a faster convergence rate and better final performance. (c) Training dynamics for different $\lambda$. A large $\lambda$ improves stability but hurts final performance, while a too-small $\lambda$ reduces stability and hinders convergence.

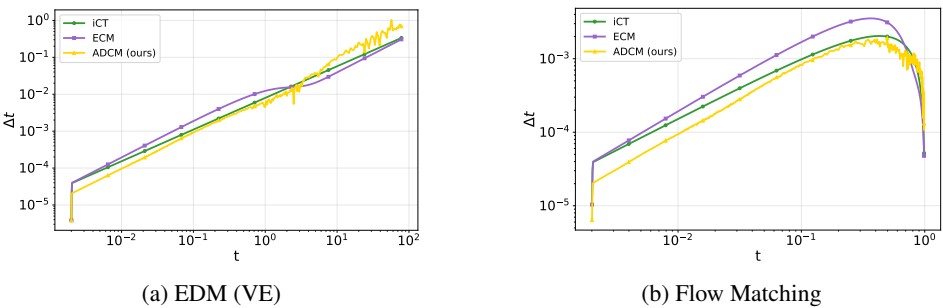

Figure 4: Adaptive Discretization on (a) EDM (VE) and (b) Flow Matching.

**Data Efficiency.** On unconditional CIFAR-10, as shown in Table 2, ADCMs achieve high-quality one-step generation results with a training budget of only 12.8M images. Compared with ECM [7], the most efficient CM to date, ADCMs achieve better generation quality with only $25\%$ of its training budget. Moreover, ADCMs surpass all previous CMs in one-step generation performance with only 76.8M training images. On class-conditional ImageNet $64 \times 64$, as shown in Table 3, ADCMs significantly reduce the training budgets of CMs. Compared to the most efficient ECM [7], ADCMs can achieve a better 1-step FID with the same model size and training budget. Moreover, ADCMs exhibit notable improvements as both the model size and training budget increase. With a $2\times$ model size, ADCMs achieve a 1-step FID of $3.49$ with a training budget of only 12.8M images. Remarkably, ADCMs are able to surpass iCT-deep [36] with only $3\%$ of its training budget.

**Computational Efficiency.** We first compare the training time cost of ADCMs with other CMs, as shown in Figure 3a. It can be observed that ADCMs introduce only about $4\%$ additional time cost under the same training epochs while improving the final performance. We also explore the convergence speed of ADCMs on unconditional CIFAR-10 with different CM approaches, as shown in Figure 3b. It can be observed that ADCMs converge significantly faster than other CM approaches, while also achieving better final performance.

## 4.2 More Results

**Adaptive Discretization Step of ADCMs.** We explore the relationship between the adaptive discretization step $\Delta t$ of ADCMs and time $t$ under different noise schedules, and compare it with existing discretization methods. We modify the discretization strategies of iCT and ECM under Flow Matching setting to be functions of $\mathrm{SNR}$ in order to enhance their performance. The results are shown in Figure 4. ADCMs are able to adaptively learn discretization strategies that are similar in trend to empirical ones without manual adjustments. In addition, compared to other discretization schemes, ADCMs adopt finer discretization at smaller $t$ and coarser discretization at larger $t$. As a result, ADCMs place greater emphasis on time intervals closer to the data during training, which aligns with empirical practices in both DMs and CMs [11, 36, 7].

Table 4: Generalization to Flow Matching. * indicates additional training costs.

| Method | NFE ($\downarrow$) | FID ($\downarrow$) |
|---|---|---|
| 1-Rectified Flow [20] | 1 | 378 |
| 2-Rectified Flow* [20] | 1 | 4.85 |
| CCM* [21] (w GAN) | 1 | 1.64 |
| Consistency-FM (w/o GAN) [43] | 2 | 5.34 |
| ECM [7] | 1 | 5.82 |
| sCT [23] | 1 | 88.52 |
| ADCM (Ours) | 1 | 5.14 |

Table 5: Scalability to ImageNet $512 \times 512$.

| Class-Conditional ImageNet $512 \times 512$ | | | |
|---|---|---|---|
| Method | Model Size | Training Budget (Mimgs) | 1-Step FID ($\downarrow$) |
| sCT | 1× | 204.8 | 10.13 |
| sCT | 2× | 204.8 | 5.84 |
| ECM | 1× | 6.4 | 25.69 |
| ECM | 2× | 6.4 | 13.55 |
| ADCM (Ours) | 1× | 6.4 | 23.12 |
| ADCM (Ours) | 2× | 6.4 | 10.53 |

$\lambda$ **as a Trade-off between Stability and Effectiveness.** We control the trade-off between the trainability and stability of ADCMs through the Lagrange multiplier $\lambda$ according to Eq. (9). We perform an ablation study on $\lambda$ by examining the training dynamics of ADCMs on unconditional CIFAR-10, as shown in Figure 3c. We find that when $\lambda$ is small, i.e., more emphasis is placed on $\mathcal{L}_{\text{global}}$, CMs converge quickly, but the final generation quality is relatively poor. When $\lambda$ is large, i.e., more emphasis is placed on $\mathcal{L}_{\text{local}}$, CMs become more unstable, making them difficult to converge and reach the optimal solution. Ablation study on loss function are deferred to Appendix B.

**ADCMs Adapt to Different Variants of DMs.** We conduct experiments on Flow Matching [18, 20], an advanced variant of DMs. We initialize CMs with pretrained DMs from [20] and compare ADCMs with other Flow Matching-based distillation methods. Additionally, we conduct experiments on ECM and sCT, two state-of-the-art CMs. All CMs are trained under a training budget of 12.8M images. As shown in Table 4, ADCMs achieve superior performance over other CMs without manual adjustments, which demonstrates the strong adaptability.

**Scalability to High-Resolution Images.** To further assess the scalability of ADCM, we conduct experiments on ImageNet $512 \times 512$. We adopt EDM2 [12] as the base latent diffusion model, which employs SD-VAE for image encoding and decoding. We compare ADCM with sCT [23] and ECM [7], the two most efficient prior CMs. The detailed results are reported in Table 5. It can be observed that ADCM scales effectively to large-scale datasets. As the model size increase, its performance improves substantially. Moreover, ADCM consistently outperforms ECM under the same training cost, further demonstrating its empirical effectiveness and training efficiency.

## 5 Conclusion

In this paper, we propose ADCMs, a unified framework for adaptive discretization in CMs. By formulating discretization as an optimization problem, we introduce local consistency as the optimization objective and global consistency as a constraint, establishing a trade-off using the Lagrange multiplier. Leveraging the Gauss-Newton method, ADCMs enable adaptive discretization, improving both trainability and stability. Experimental results show that ADCMs significantly improve training efficiency and final performance of CMs on different datasets while demonstrating strong adaptability to different variants of DMs.

## Acknowledgments and Disclosure of Funding

Z. Ling is partially supported by the National Natural Science Foundation of China (via NSFC-62406119), the Natural Science Foundation of Hubei Province (2024AFB074), and the Guangdong Provincial Key Laboratory of Mathematical Foundations for Artificial Intelligence (2023B1212010001). Z. Deng is partially supported by the National Natural Science Foundation of China (via NSFC-92470118 and NSFC-62306176) and the Natural Science Foundation of Shanghai (23ZR1428700). R. C. Qiu is partially supported in part by the National Natural Science Foundation of China (via NSFC-12141107), the Key Research and Development Program of Wuhan (2024050702030100), and the Key Research and Development Program of Guangxi (GuiKe-AB21196034).

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

# A  Experiments Details

## A.1  Precondition

For VE-based ADCMs, we follow the parameterization of EDM. Specifically, we set:

$$c_{\text{skip}}(t) = \frac{\sigma_{\text{data}}^2}{\sigma_{\text{data}}^2 + t^2},\ c_{\text{out}}(t) = \frac{\sigma_{\text{data}}t}{\sqrt{\sigma_{\text{data}}^2 + t^2}},\ c_{\text{in}}(t) = \frac{1}{\sqrt{\sigma_{\text{data}}^2 + t^2}},\ c_{\text{noise}}(t) = \frac{1}{4}\log t.$$

For ADCMs on the Flow Matching setting, using EDM's precondition causes the model output to deviate from $x_0$, which contradicts the objective of CMs to estimate $x_0$. We use a pretrained model from rectified flow [20] whose output is:

$$F_\theta(x_t, t) = \frac{x_t - x_0}{t},$$

which implies $c_{\text{in}}(t) = 1$ and $c_{\text{noise}}(t) = t$. To ensure the model's final output matches $x_0$, we accordingly modify the preconditioning to:

$$c_{\text{skip}}(t) = 1, \quad c_{\text{out}}(t) = -t.$$

## A.2  Hyperparameters

**Batch Size and EMA.**  For unconditional CIFAR-10, we use a batch size of $128$ with an EMA decay rate of $0.9999$ for training budget of $12.8$M images. We use a batch size of $1024$ with an EMA decay rate of $0.99993$ for training budget of $76.8$M images. For class-conditional ImageNet $64 \times 64$, we set the batch size to $128$, $256$, and $512$, corresponding to training budgets of $12.8$M, $25.6$M, and $51.2$M images, respectively. We use Power function EMA for class-conditional ImageNet $64 \times 64$ following [12].

**Time Sampling.**  For unconditional CIFAR-10, we follow previous works [36, 7, 23] and use a log-normal SNR distribution for time sampling, which can be expressed as:

$$\log(\text{SNR}(t)) \sim \mathcal{N}(P_{\text{mean}}, P_{\text{std}}^2)$$

where $\text{SNR}(t) = \frac{\beta_t}{\alpha_t}$, $P_{\text{mean}} = -1.1$, $P_{\text{std}} = 2.0$. Since the time segment $\mathbb{T}$ is discrete, we also apply discretization to the sampling results following [36]. For class-conditional ImageNet $64 \times 64$, we sample uniformly within the time segment $\mathbb{T}$.

**Lagrange Multiplier $\lambda$.**  For unconditional CIFAR-10, we set $\lambda = 0.01$. For class-conditional ImageNet $64 \times 64$, we find that starting with a small $\lambda$ led to training instability on ImageNet $64 \times 64$. Therefore, we follow previous work [36, 7] and select $\lambda = 0.64$ for warm-up, gradually decreasing it to $\lambda = 0.01$. We summarize the hyperparameter settings in Table 6.

## A.3  Baseline Reimplementation.

Some of the baseline results in this paper, including those in Figure 3b, Table 4 and sCM [23] under VE settings, are obtained from our own reproductions. Under the VE SDE setting, for a fair comparison, we initialize all neural networks using the pretrained DM provided by EDM [11]. We also adopt a consistent EMA decay rate of $0.9999$ and a dropout probability of $30\%$ (except for CD where dropout is set to $0$, as dropout can lead to a decline in CD's performance). We do not make further modifications to other parameters. For sCM under the VE SDE and Flow Matching setting, we apply the advanced training techniques from [23], except for the network architecture changes, allowing sCM to utilize pretrained DMs. We do not use the adaptive weighting and tangent warmup techniques, as we find that they degrade the performance of sCM. For all baselines under the Flow Matching setting, we replace their original discretization scheme, time sampling, and weighting function—from being functions of time $t$ to being functions of SNR. It is important to note that without manual adjustments, the performance of these baselines degrades significantly (e.g., the 1-step FID of ECM drops from $5.82$ to $15.55$). The implementation code is available in the supplementary material.

Table 6: Hyperparameter Settings

| | Unconditional CIFAR-10 | Class-conditional ImageNet $64 \times 64$ | |
|---|---|---|---|
| Base model | EDM [11] | EDM2-S [12] | EDM2-M [12] |
| Model capacity (Mparams) | 55.7 | 280.2 | 497.8 |
| Model complexity (GFLOPS) | 21.3 | 101.9 | 180.8 |
| GPU types | RTX3090 | RTX3090 | A100 |
| GPU memory | 24G | 24G | 40G |
| Number of GPUs | 1 | 8 | 4 |
| Dropout probability | 30% | 40% | 50% |
| Optimizer | RAdam | Adam | Adam |
| Learning rate schedule | fixed | square root | square root |
| Learning rate max | 0.0001 | 0.001 | 0.0009 |
| Pseudo-Huber c | 0.03 | 0 | 0 |
| Time sampling | log-normal SNR | uniform | uniform |
| $P_{\text{mean}}$ | -1.1 | - | - |
| $P_{\text{std}}$ | 2.0 | - | - |

# B   Ablation Study

We investigate the impact of adaptive loss function in ADCMs, including the choice of weighting function and distance metric. We perform an ablation study on unconditional CIFAR-10 under the same training budget of 12.8M images.

Table 7: Ablation Study on Weighting Function.

| Weighting Function | 1-Step FID ($\downarrow$) |
|---|---|
| 1 | 5.70 |
| $\frac{1}{t_i - t_{i-1}}$ | 4.09 |
| $\frac{1}{t_i}$ | 3.84 |
| Adaptive weighting (Ours) | **3.16** |

**Weighting Function.**   The choice of weighting function is crucial for training CMs. An inappropriate weighting function can lead to imbalanced optimization over time, ultimately degrading performance. We investigate the impact of different weighting functions on ADCMs. The detailed results are presented in Table 7. The results show that our designed adaptive weighting function can effectively enhance the generation capability of ADCMs. Notably, even without the loss function improvement, ADCMs still outperform ECM's 1-step FID (4.54). This demonstrates that the improvement of ADCMs mainly comes from our designed adaptive discretization strategy while our proposed adaptive weighting function further enhances the performance of ADCMs.

**Distance Metric.**   We investigate the effect of different distance metrics on the performance of ADCMs. Following common practice in prior works [7, 36, 17], we adopt the Pseudo-Huber metric due to its robustness to outliers [36]. The Pseudo-Huber metric is defined as

$$d(\boldsymbol{x}, \boldsymbol{y}) = \sqrt{\|\boldsymbol{x} - \boldsymbol{y}\|_2^2 + c^2} - c,$$

which provides a smooth interpolation between the $L_2$ and squared $L_2$ metrics. Specifically, when $c = 0$, it reduces to the standard $L_2$ distance, while as $c \to \infty$, it approaches the squared $L_2$ distance. smoothly bridges the gap between the $L_2$ and squared $L_2$ metric. When $c = 0$, the Pseudo-Huber metric degenerates to the standard $L_2$ metric. When $c \to \infty$, it becomes equivalent to the squared $L_2$ metric. To address this phenomenon, we conduct experiments with different values of $c$, and the

Table 8: Ablation Study on Distance Metric.

| Distance Metric | 1-Step FID ($\downarrow$) |
|---|---|
| $L_2$ | 3.54 |
| Pseudo-Huber (c=0.0003) | 3.44 |
| Pseudo-Huber (c=0.003) | 3.42 |
| Pseudo-Huber (c=0.03) | **3.16** |
| Pseudo-Huber (c=0.3) | 4.42 |
| Pseudo-Huber (c=3) | 5.23 |
| Squared $L_2$ | 5.33 |

results are presented in Table 8. It can be observed that Pseudo-Huber metric smoothly interpolates between $L_2$ and squared $L_2$ through the control of the parameter $c$, thus achieving performance that surpasses both extremes. These results clearly demonstrate the significance of choosing Pseudo-Huber as our distance metric.

We also examine the impact of mismatched distance metrics between the original CM loss function and the weighting function. We fix the distance metric applied to the original CM loss function as Pseudo-Huber with $c = 0.03$, while applying different distance metrics in the weighting function. As shown in Table 9, a mismatched distance metric leads to degraded performance of ADCMs.

Table 9: Impact of Mismatched Distance Metric.

| Distance Metric on Weighting Function | 1-Step FID ($\downarrow$) |
|---|---|
| Squared $L_2$ | 4.09 |
| $L_2$ | 3.36 |
| Pseudo-Huber (c=0.03) | **3.16** |

## C   Two Step Generation

Compared to other distillation methods for DMs, CMs have the significant advantage of preserving the inherent characteristics of DMs, specifically, the ability to improve generation quality through multi-step sampling. We investigate the two-step generation performance of ADCMs on unconditional CIFAR-10, with results shown in Table 10. We set the intermediate $t = 0.420$. It can be observed that ADCMs not only maintain optimal single-step generation performance but also demonstrate strong two-step generation capabilities, second only to ECM [7], which is specifically designed for two-step generation.

Table 10: 2-step generation results on unconditional CIFAR-10. ADCMs achieve the best 1-step FID while also attaining the second-best 2-step FID.

| Method | 1-Step FID ($\downarrow$) | 2-Step FID ($\downarrow$) |
|---|---|---|
| iCT | 4.18 | 2.58 |
| sCM (VE) | 5.62 | 2.73 |
| ECM | 4.54 | **2.20** |
| ADCM (Ours) | **3.16** | 2.44 |

## D   Limitations and Broader Impacts

In this paper, we introduce ADCMs, an adaptive discretization method for CMs. Our approach effectively improves both the training efficiency and generation quality of CMs, and demonstrates adaptability to different variants of DMs. However, ADCMs focus on Consistency Training (CT), as it generally yields better performance. In the case of Consistency Distillation (CD), estimating

$\mathcal{L}_{\text{global}}$ significantly increases training costs due to the need for iterative solving of the endpoint of the PF-ODE. We leave this issue for future work. ADCMs enable efficient content generation for creators while significantly reducing the computational cost of obtaining similar models. Additionally, similar to other deep generative models, ADCMs could be misused to generate harmful fake content, and the proposed method may further exacerbate the potential risks associated with malicious applications of such models.

# E  License

We list the used datasets, models and their citations, URLs, and licenses in Table 11.

Table 11: Licenses and citations for existing assets.

| Name | URL | Citation | License |
|---|---|---|---|
| CIFAR-10 | `https://www.cs.toronto.edu/~kriz/cifar.html` | [16] | \ |
| ImageNet | `https://www.image-net.org` | [3] | \ |
| EDM | `https://github.com/NVlabs/edm` | [11] | Creative Commons Attribution-NonCommercial-ShareAlike 4.0 International License |
| EDM2 | `https://github.com/NVlabs/edm2` | [12] | Creative Commons Attribution-NonCommercial-ShareAlike 4.0 International License |
| Rectified Flow | `https://github.com/gnobitab/RectifiedFlow` | [20] | \ |

# F  Additional Samples

We provide additional samples of ADCMs from unconditional CIFAR-10 and class-conditional ImageNet $64 \times 64$ in Figures 5 - 7.

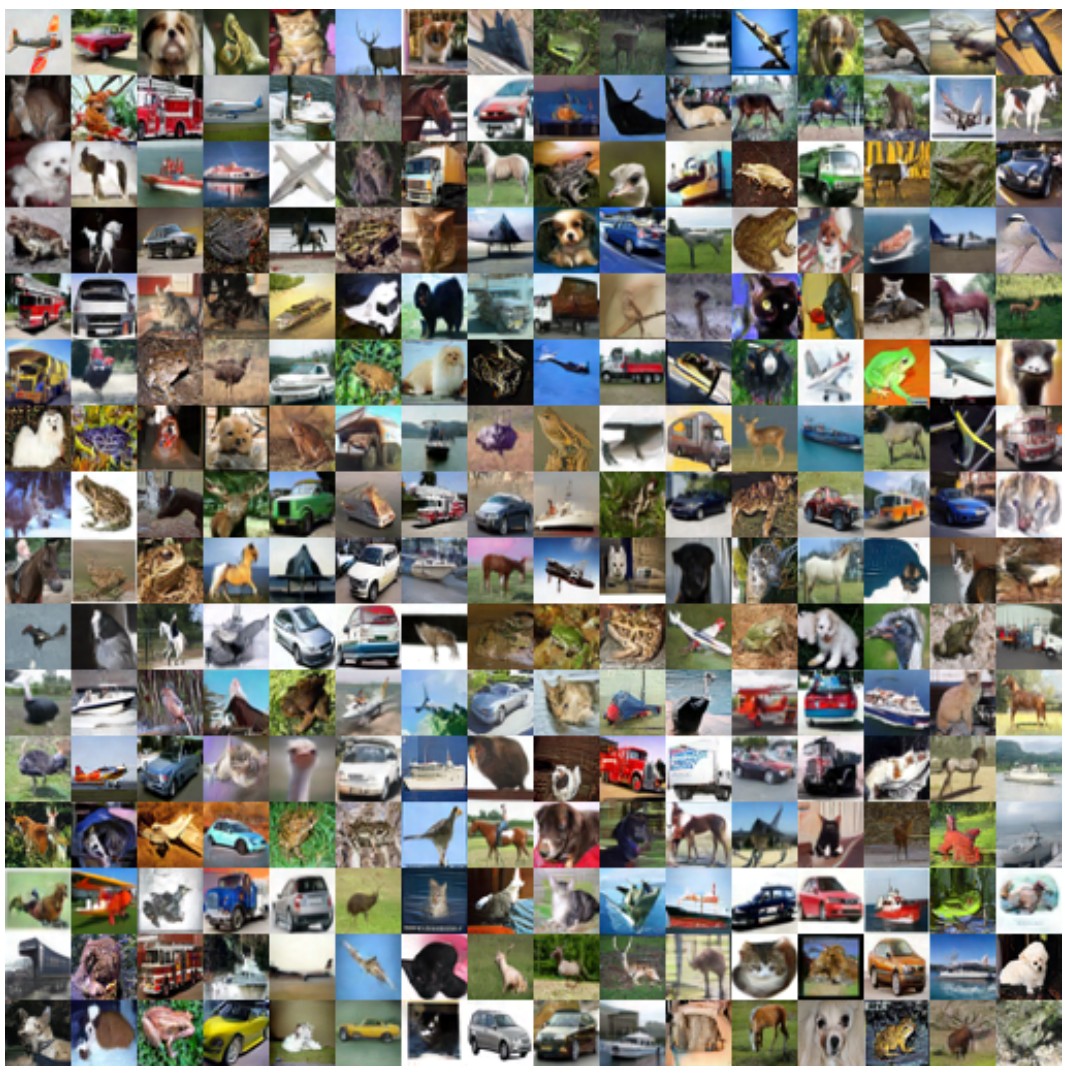

Figure 5: 1-step samples from unconditional CIFAR-10 trained with a budget of 76.8M images.

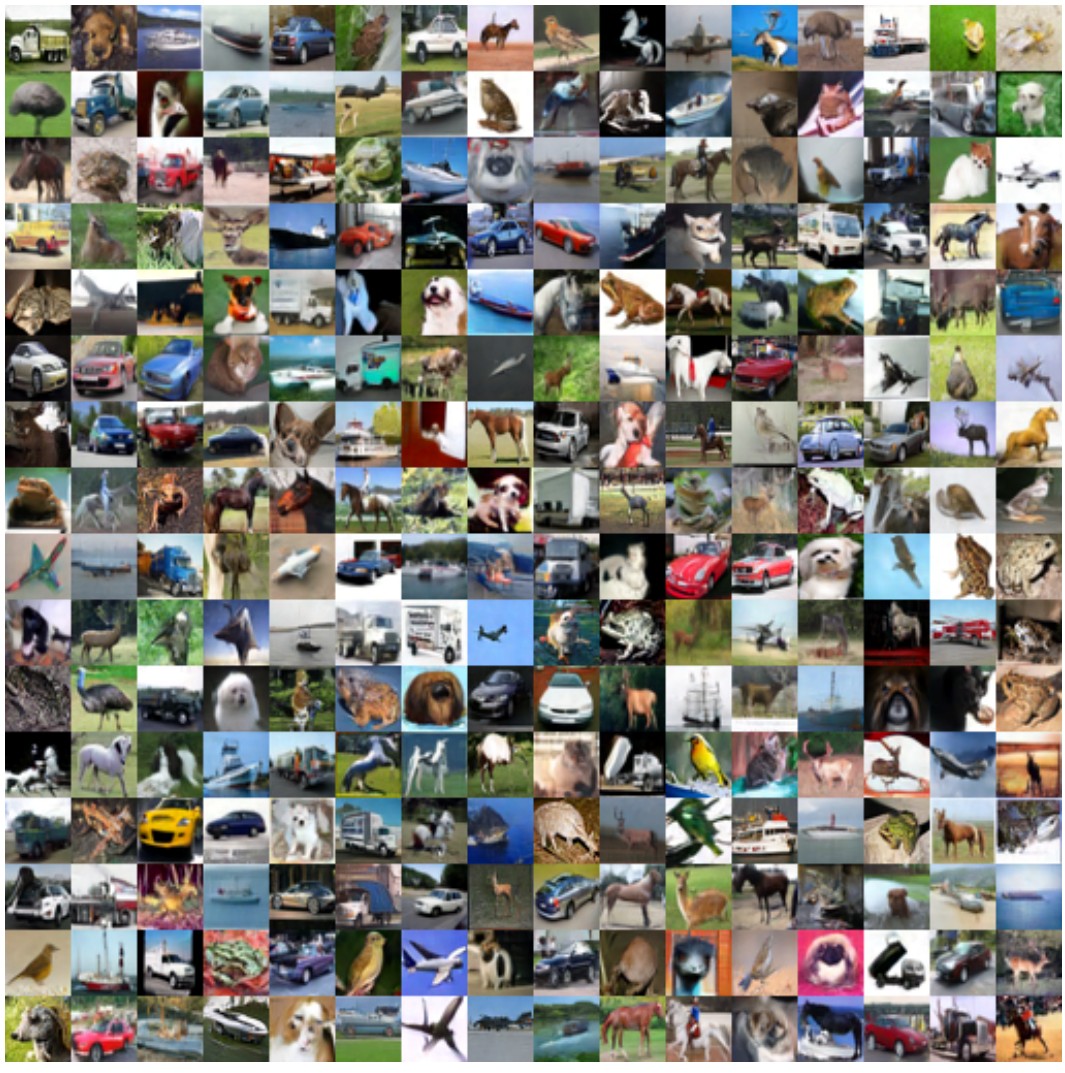

Figure 6: 1-step samples from unconditional CIFAR-10 trained with a budget of 12.8M images.

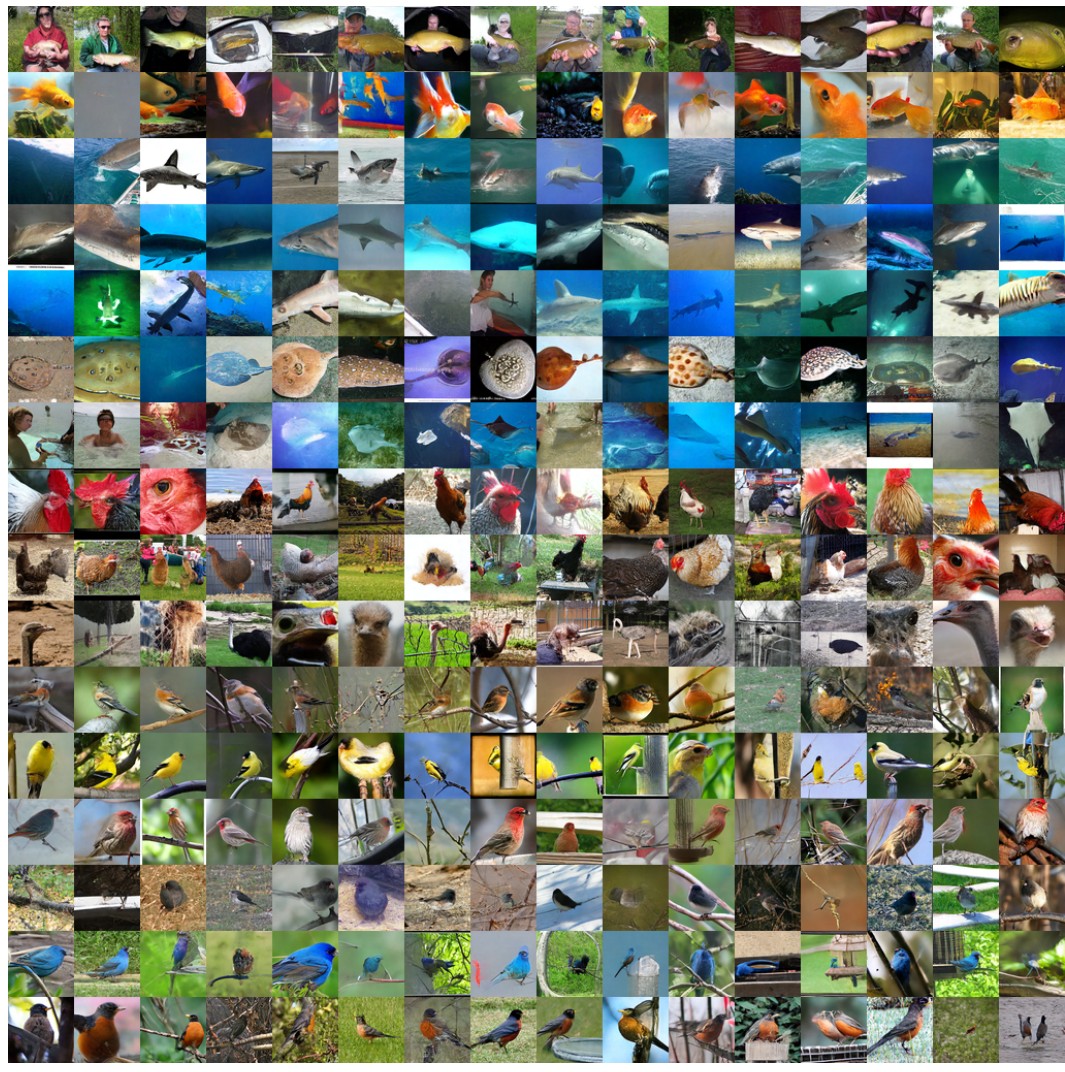

Figure 7: 1-step samples from class-conditional ImageNet $64 \times 64$ trained with a budget of 12.8M images.

