# OpenReview forum: "Adaptive Discretization for Consistency Models"
_NeurIPS.cc/2025/Conference — NeurIPS 2025 poster_

### Official Review · Reviewer_TPUd · 2025-06-09

**Clarity:** 3
**Significance:** 2
**Originality:** 2
**Rating:** 4
**Confidence:** 4

**Summary:**

The paper introduces a form of adaptive time discretization of consistency training designed to replace the fixed discretization schedules that are commonly adopted in the literature. The choice of the time step Dt is obtained as the result of an optimization of a linear mixture the standard consistency loss and a denoising autoencoder loss. This choice is interpreted in terms of a trade-off between local error (consistency) and global error. The proposed method seems to work well in practice, especially on CIFAR10 where it achieves SOTA performance. The method oslo seems to have an advantage for smaller dataset sizes, although it is unclear if this advantage would scale in the large data regime.

**Questions:**

Can you explain to me why you are interpreting Eq.6 as the global accumulated error of the consistency model? (see above).

The correct expression for the true global error should be:

global_error = E_{x_0}||  f(g_t(x_0)) - x_0 ||^2

where g_t(x_0) is the ODE flow. This expression is zero when f = g^{-1}, which is the optimum of a consistency model. On the other hand, the expression

denoising_error = E_{x_0, x_t}||  x_t - x_0 ||^2

is only exactly equal to zero when p(x_0 | x_t) is a delta distribution.

**Ethical Concerns:**

["NO or VERY MINOR ethics concerns only"]

**Final Justification:**

The methodological approach is solid and the results are promising. Some parts of the theoretical explanation need to be revised based on our discussion.

**Limitations:**

There is no "Limitation' section in the main text and the section in the Supps. is very limited. I would have appreciated a more extensive discussion, especially since it is not clear if the proposed approach would scale well to the large data regime.

**Paper Formatting Concerns:**

Everything looks fine

**Quality:**

2

**Strengths And Weaknesses:**

I think that this is a very important problem area that might drive consistency training to generative SOTA. The standard discretization approaches are largely handcrafted and often ignore the rich mathematical structure that is implicit in these models. I do think that  the algorithm proposed by the authors is an important step in the right direction. I find the results to be very exciting since it is notoriously hard to improve on the very optimized iCT baseline.
I really appreciate the use of principled closed-form and Taylor methods to obtain the ‘optimal’ schedule, which is a neat example of interplay between theory and practical machine learning.

My main issue with this work is on the interpretation of the main constraint is Eq.6. As far as I understood it, the authors characterize the expression as the global error due to the accumulation of training errors along the ODE flow trajectory. However, this interpretation is just wrong since, even when the network is perfectly trained, the denoising error given in the constraint will be generally non-zero.
The reason is that, if the noise is sampled independently under the forward process, the state x_t will generally not lie on the same ODE trajectory as x_0. In fact, the optimum of the denoising loss can be expressed in terms of the score function instead of the consistency map.
So said, the denoising autoencoder loss is a central quantity in both diffusion and consistency, so it can well be very meaningful to have it as a constraint.

---

> ### Author Rebuttal · Authors · 2025-07-31
>
> Thank you very much for your encouraging feedback, your appreciation of our theoretical and practical contributions, and your deep understanding of this important problem. Below, we provide point-by-point responses to your concerns and questions.
>
> **Global Error:** We first emphasize that ADCM focuses on the Consistency Training (CT) paradigm[^1] due to its superior empirical performance[^2][^3][^4][^5], as noted in line 38. Under CT paradigm, ADCM has access to the data point $x_0$, which allows it to obtain **an unbiased score estimation $s(x, t) = \frac{x - \alpha_t x_0}{\beta_t^2}$**, as described in the original CM paper[^1]. Under this assumption，$x_t = \alpha_t x_0 + \beta_t z$ and $x_0$ lie on the same Probability Flow ODE (PF-ODE) trajectory for $\forall t \geq0$. For simplicity, we take the VE setting as an example. At this point, $x_t$ is given by $x_t = x_0 + tz$, and PF-ODE is given by
>
> $\frac{dx(r)}{dr} = -r \cdot s(x(r),r) \quad \to \quad \frac{dx(r)}{dr} = \frac{x_0 - x(r)}{r}$.
>
> We will prove $x_t = x_0 + tz$ and $x_0$ lie on the same PF-ODE trajectory from two perspectives.
>
> 1. We first consider the standard PF-ODE sampling process, i.e., taking $x(t) = x_t = x_0 + t z$ as the initial condition and solving the PF-ODE starting from time $t$.
>
> In this case, the solution to the PF-ODE is given by $x(r) = x_0 + r z$. Substituting $r = 0$ yields the original data point $x_0$, confirming that $x_t$ and $x_0$ lie on the same PF-ODE trajectory.
>
> 2. To further address your concern, we consider the global error proposed by the reviewer. Under the CT paradigm, the ODE flow $g_t(x_0)$ mentioned by the reviewer corresponds to the solution $x(t)$ of the PF-ODE with initial condition $x(0) = x_0$, solved forward from $r = 0$ to time $t$.
>
> In this case, the PF-ODE can only be solved under the assumption that $\lim_{r \to 0} \frac{x_0 - x(r)}{r} = z$, where $z$ is an arbitrary vector. Under this condition, the solution to the PF-ODE is $x(r) = x_0 + r z$. By taking $r = t$, we obtain $g_t(x_0) = x(t) = x_0 + t z$. Therefore, under the CT assumption, our definition of the global error is equivalent to that proposed by the reviewer.
>
> The denoising error proposed by the reviewer does not involve any NN as it directly measures the distance between the noised data $x_t$ and the original data $x_0$. As such, its meaning is not entirely clear to us. We would greatly appreciate it if the reviewer could provide further clarification on the denoising error. We would be pleased to address any questions or concerns the reviewer may have.
>
> **Large Scale Results:** To address your concern, we conduct experiments on ImageNet at $512 \times 512$ resolution. We choose EDM2[^6] as the base latent diffusion model which employ SD-VAE for image encoding and decoding. The detailed results are presented in the following table. We choose to compare our method with ECM[^2] and sCT[^3], the most efficient prior CM, and our results are promising. We outperform ECM,  with a similar training budget. While we do not yet match sCT’s performance, their method involves extremely higher training costs  and specialized architecture. The results demonstrate that ADCM is capable of scaling to high-resolution images while maintaining high efficiency.
>
> | Method | Training Budgets (Mimgs) | 1-step FID |
> |:---:|:---:|:---:|
> |sCT-S                  |204.8|  10.13 |
> |ECM-S             |  6.4 | 25.69 |
> |ADCM-S (Ours)          |  6.4 | 23.12 |
>
> [^1]:Song Y, et al. Consistency Models. ICML 2023.
> [^2]:Geng Z, Pokle A, Luo W, et al. Consistency models made easy. ICLR 2025.
> [^3]:Song Y, et al. Improved Techniques for Training Consistency Models. ICLR 2024.
> [^4]:Lu C et al. Simplifying, stabilizing and scaling continuous-time consistency models. ICLR 2025.
> [^5]:Lee S, Xu Y, Geffner T, et al. Truncated consistency models. ICLR 2025.
> [^6]:Karras T, Aittala M, Lehtinen J, et al. Analyzing and improving the training dynamics of diffusion models. CVPR 2024.

---

> ### Comment · Reviewer_TPUd · 2025-08-03
> **Incorrec t understanding of PDF-ODE paths**
>
> Dear authors,
>
> Thank you for the reply. Unfortunately, I must point out that you have an incorrect understanding of consistency models and PDF-ODE trajectories.
>
> It is completely obvious that the PDF-ODE path cannot be $\gamma(t, x_0) = x_0 + t z$ sampled under the forward process since $z$ is sampled independently from $x_0$ and can have any value, while the PDF-ODE path $\gamma(t, x_0) $ is defined uniquely and has a well-defined terminal value \gamma(T, x_0)  (the noise deterministically associated with the image x_0).
>
> In the general case, the PDF-ODE path $\gamma(t, x_0) $ is not a straight line and it cannot be expressed as $\gamma(t, x_0)  = x_0 + t z^* $ for any noise vectors $z^*$. It is true that $\frac{x_0 - x(t)}{\sigma^2(t)}$ is un unbiased estimator of the score, but this would give you an unbiased stochastic dynamics only if you re-sample $x_0$ for any given $x(t), t$ under the probability $p(x_0 \mid x(t),t)$. In any case, we are considering the deterministic dynamics here, which requires the exact score, not just an unbiased one-sample estimator of it.
>
> In fact, the global denoising error that you are considering is identical to the consistency error if and only if $\gamma(t, x_0)$ is a straight line, which in a standard consistency model is true if and only if the dataset has a single datapoint. In this case, the mapping is not 1-to-1 and all arbitrary noise values map to the same x_0, making the linear equation correct. Needless to say, this case is not representative of real datasets and it is only sometimes valid approximately in the low noise regime.

---

> > ### Author Response · Authors · 2025-08-04
> >
> > **Clarification on the Interpretation of the Constraint:** Thank you for your thoughtful comment and detailed explanation. We realize that we initially misunderstood your point. We agree that the denoising error in Eq.(6) does not vanish even for a perfectly trained model due to the stochasticity of the forward process. Nonetheless, our constraint can be viewed as a tractable surrogate that promotes consistency across samples from the same underlying distribution as noted in line 176, aligning with the spirit of consistency training. We appreciate your insightful observation that our constraint closely relates to the denoising autoencoder loss, a central quantity in both diffusion and consistency models. We believe this connection offers a unifying perspective and will clarify it in the final version to avoid potential confusion. We also **sincerely** appreciate your deep understanding of the underlying principles and your constructive feedback, which helped us better frame our contributions.
> >
> > **Updated High-Resolution Experiments:** We update the experimental results on high-resolution images with increased training budget and larger model, as shown in the table below. "S" stands for small with model capacity of 280.2M parameters and "M" stands for middle with model capacity of 497.8M parameters, following the notation used in sCT[^1]. It can be observed that our method scales well on large datasets. This further demonstrates the empirical effectiveness of our method. We are currently conducting experiments on larger scales. While these experiments are progressing well, full results are pending due to time and computational resource constraints. We will continue to update the results as they become available.
> > | Method | Training Budgets (Mimgs) | 1-step FID |
> > |:---:|:---:|:---:|
> > |sCT-S                  |204.8|  10.13 |
> > |sCT-M                  |204.8|  5.84 |
> > |ECT-S                  | 6.4 |  25.69 |
> > |ECT-S                  | 12.8 |  18.51 |
> > |ECT-M                  | 6.4 |  13.55 |
> > |ADCM-S (Ours)          | 6.4 | 23.12 |
> > |ADCM-S (Ours)          | 12.8 | 17.24 |
> > |ADCM-M (Ours)          | 6.4 | 10.53 |
> >
> > We appreciate your time and effort in evaluating our work and look forward to any feedback you may have.
> >
> > [^1]:Lu C et al. Simplifying, stabilizing and scaling continuous-time consistency models. ICLR 2025.

---

> > > ### Comment · Reviewer_TPUd · 2025-08-05
> > >
> > > Thank you for the acknowledgment and for the new results. I think that your work is valuable and I am happy to keep my original score. It would be useful if you could incorporate some of our discussion into the text to avoid confusions.

---

### Official Review · Reviewer_nhUt · 2025-07-03

**Clarity:** 3
**Significance:** 3
**Originality:** 3
**Rating:** 4
**Confidence:** 4

**Summary:**

This paper presents a technique for adaptively choosing the discretization steps in consistency models to improve their training stability and efficiency. The idea is to formulate the problem using a combination of local and global consistency losses. Using Gauss-Newton method, the authors derive an analytical solution to the problem. This results in a formulation where \delta_t can be derived using JVP. Using this, the authors show that superior one step generation results can be obtained with much better training efficiency on Imagenet64 and CIFAR-10 datasets.

**Questions:**

1. For large-scale runs, how would you recommend picking \lambda?
2. From table 1, it looks like ICT or TCM still outperform ADCM. With more training budget, can you match or outperform these methods?

**Ethical Concerns:**

["NO or VERY MINOR ethics concerns only"]

**Final Justification:**

The paper addressed my concerns about large scale experiments. It is promising to see that results on Imagenet with -M model gets good FID. The paper has good contribution, hence I stay with my initial rating of Borderline accept.

**Limitations:**

Yes

**Paper Formatting Concerns:**

-

**Quality:**

3

**Strengths And Weaknesses:**

Strength:
1. The paper addresses an important problem of choosing discretization steps in consistency models.
2. The proposed method is novel. The formulation is intuitive, the objective makes sense. Overall, the results are easy to follow.
3. Experimental results are very strong. I am impressed with the training efficiency section/

Weakness:
1. The paper would have been very strong if there were large-scale results on high resolution with LDMs. I would not penalize the authors on this aspect as some labs can have low resources.
2. Among other consistency models, results are still not SOTA. Other methods can outperform ADCM.

---

> ### Author Rebuttal · Authors · 2025-07-31
>
> Thank you for your positive evaluation and kind words regarding our method’s novelty, intuitive formulation, and strong experimental results. We also appreciate your recognition of the training efficiency improvements and address your concerns below.
>
> **Higher Resolution Generation Results:** Following your suggestion, we conduct experiments on ImageNet at $512 \times 512$ resolution. We choose EDM2[^1] as the base latent diffusion model, which employs SD-VAE for image encoding and decoding. The detailed results are presented in the following table.
> Training at this scale is time-consuming, which limits the extent of experiments we can perform during the rebuttal period. Moreover, we note that only sCT[^2] reports results at this resolution. Comprehensive comparisons with other baselines are difficult, as training and re-implementing these methods demand significant time. Thus, we choose to compare our method with sCT and ECM, the most efficient prior CM.
> Our results are promising. We outperform ECM[^3],  with a similar training budget. While we do not yet match sCT’s performance, their method involves extremely higher training costs  and specialized architecture. The results demonstrate that ADCM is capable of scaling to high-resolution images while maintaining high efficiency.
>
> | Method | Training Budgets (Mimgs) | 1-step FID |
> |:---:|:---:|:---:|
> |sCT-S                  |204.8|  10.13 |
> |ECM-S             |  6.4 | 25.69 |
> |ADCM-S (Ours)          |  6.4 | 23.12 |
>
> **$\lambda$ for Large Scale Runs:** Due to the inherent instability of models in large-scale experiments, we recommend using a larger $\lambda$ to ensure training stability or just employ a larger $\lambda$ in the early training stage as a warm-up. This recommendation aligns with the design of our theoretical framework: a larger  $\lambda$  places greater emphasis on global consistency, thereby enhancing stability.
>
> [^1]:Karras T, Aittala M, Lehtinen J, et al. Analyzing and improving the training dynamics of diffusion models. CVPR 2024.
> [^2]:Lu C et al. Simplifying, stabilizing and scaling continuous-time consistency models. ICLR 2025.
> [^3]:Geng Z, Pokle A, Luo W, et al. Consistency models made easy. ICLR 2025.

---

> ### Author Response · Authors · 2025-08-04
> **Updated Additional Experiments**
>
> **Updated High-Resolution Experiments:** We update the experimental results on high-resolution images with increased training budget and larger model, as shown in the table below. S stands for small with model capacity of 280.2M parameters and M stands for middle with model capacity of 497.8M parameters, following the notation used in sCT[^1]. It can be observed that our method scales well on large datasets. This further demonstrates the empirical effectiveness of our method. We are currently conducting experiments on larger scales. While these experiments are progressing well, full results are pending due to time and computational resource constraints. We will continue to update the results as they become available.
> | Method | Training Budgets (Mimgs) | 1-step FID |
> |:---:|:---:|:---:|
> |sCT-S                  |204.8|  10.13 |
> |sCT-M                  |204.8|  5.84 |
> |ECT-S                  | 6.4 |  25.69 |
> |ECT-S                  | 12.8 |  18.51 |
> |ECT-M                  | 6.4 |  13.55 |
> |ADCM-S (Ours)          | 6.4 | 23.12 |
> |ADCM-S (Ours)          | 12.8 | 17.24 |
> |ADCM-M (Ours)          | 6.4 | 10.53 |
>
> [^1]:Lu C et al. Simplifying, stabilizing and scaling continuous-time consistency models. ICLR 2025.

---

### Official Review · Reviewer_gb53 · 2025-07-06

**Clarity:** 3
**Significance:** 2
**Originality:** 3
**Rating:** 4
**Confidence:** 3

**Summary:**

Existing works on consistency models often rely on manually designed or ad-hoc discretization schemes, which can be inefficient and lead to poor training outcomes. This paper introduces ADCMs, a unified framework that addresses this by automatically and adaptively determining the discretization strategy. The method formulates discretization as a constrained optimization problem, using local consistency—minimizing the error between adjacent trajectory points—as the primary objective to ensure the model is trainable. Simultaneously, it imposes a constraint on global consistency—controlling the accumulated error relative to the ground-truth data—to maintain training stability and prevent divergence. To effectively balance this trade-off between trainability and stability, the authors employ a Lagrange multiplier and leverage the Gauss-Newton method to derive an efficient, analytical solution for the optimal discretization step. Several experiments are conducted to demonstrate that the proposed method, ADCMs, improve the efficiency of CMs.

**Questions:**

- In eqn 2, why we optimize over f_\theta(x_t) rather than f_\theta- (x_t)
- In eqn 6, why the bound \delta is the same over all t, i.e., why don't you use \delta_t ?

**Ethical Concerns:**

["NO or VERY MINOR ethics concerns only"]

**Final Justification:**

I raise my score from 3 to 4 as the author addressed most of my concerns in the rebuttal.

**Limitations:**

Yes

**Quality:**

2

**Strengths And Weaknesses:**

Strengths:
- The paper is well-written and easy to follow. All math notations are defined and explained before being used.
- The proposed optimization objective has a clear motivation and is easy to implement.
- Several experiments and ablation studies are conducted to demonstrate the performance of the proposed method.


Weaknesses:
- line 173: please explain further on why you should optimize over f_\theta- (x_t-\delta t) rather than  f_\theta- (x_t)
- The function stopgrad() is not defined before usage
- The computational complexity of the algorithm 1 is not analyzed. The paper also lacks the analysis of the convergence rate.
- The approximation error of the methods in Sec 3.2 is not discussed
- The experiments only adopt CIFAR-10 and ImageNet datasets, which are very limited. More advanced datasets should be adopted.
- The ablation studies should cover more values of \lambda (\lambda = \infty)

---

> ### Author Rebuttal · Authors · 2025-07-31
>
> We thank the reviewer for your careful consideration. We greatly appreciate the positive comments and address your concerns below.
>
> **Optimization over $f_{\theta^-}(x_t-\Delta t)$:** We kindly note that our objective is to enforce global consistency by optimizing $\Delta t$. Global consistency is reflected in **the cumulative error between the practical training target $f_{\theta^-}(x_t - \Delta t)$ in Eq.(2) and the actual target $x_0$ (i.e., the data)**. This implies that, when $\Delta t = 0$, the practical target becomes $f_{\theta^-}(x_t)$, which often leads to a larger cumulative error. In this case, global consistency is at its weakest, resulting in unstable training. Therefore, we choose to optimize $f_{\theta^-}(x_t-\Delta t)$ instead of $f_{\theta^-}(x_t)$.
>
> **Stopgrad Definition:** $\operatorname{stopgrad}(\theta)$ function is is a standard component in  previous Consistency Model works[^1][^2][^3][^4]. To address your concern, we will clarify its definition and explain why $\theta^-$ can't be optimized after applying $\operatorname{stopgrad}$. It treats the given parameter $\theta$ as a constant during gradient computation. This means that while the forward pass still uses the actual value, the backward pass cuts off the gradient flow.   As a result, we cannot optimize over $f_{\theta^-}(x_t)$ since it does not participate in gradient computation.
>
>
> **Computational Complexity of Algorithm 1:** The computational overhead in Algorithm 1 compared to other methods comes from the computation of $\mathbb{T}$. For each value of $\Delta t$, we use a mini-batch to estimate the expectation and compute the Jacobian. The computational overhead introduced by the Jacobian calculations is negligible. As noted in line 224, the Jacobian can be efficiently computed using **PyTorch’s built-in Jacobian-vector product (JVP)**.
>
> Moreover, since we adopt the Gauss-Newton method, which incorporates second-order approximation information, the optimization typically converges faster than first-order methods[^5], further reducing the overall computational cost. Additionally, due to the infrequent updates of $\mathbb{T}$, the overhead of computing $\mathbb{T}$ is minimal relative to the total training time.
>
> As shown in Figure 3a, the additional computational cost is typically less than 4% under the same number of training epochs, while significantly improving performance. As shown in Figure 3b, ADCM significantly accelerates the convergence of Consistency Models (CM), allowing it to achieve better generative performance with fewer training steps compared to other CMs.
>
> **Approximation Error**: To solve the optimization problem in Eq.(8), we adopt the Gauss-Newton method, as it is well-suited for problems involving complex neural networks. Specifically, we approximate $f_{\theta^-}(x_t - \Delta t)$ using the first-order Taylor expansion of the neural network. It is important to note that this approximation error is acceptable, since, as stated in line 265, we do not require precise step sizes but only need to capture the trend of step size changes over time $t$. To further address your concern, we calculated the average approximation error incurred during the computation of $\mathbb{T}$ with different $\lambda$. The error is measured by $\frac{\|f_{\theta^-}(x_{t-\Delta t}) - (f_{\theta^-}(x_{t}) - v \Delta t) \|}{\| f_{\theta^-}(x_{t-\Delta t}) \|}$. The detailed results are presented in the following table. It can be observed that the approximation error is very small even with a large $\lambda$, demonstrating that the Gauss-Newton method provides a sufficiently accurate analytical solution in our setting.
>
> | $\lambda$ | Approximation Error |
> |:---:|:---:|
> |0.64           |0.0357 |
> |0.16           |0.0050 |
> |0.04           |0.0004 |
> |0.01          |0.00002|
>
>
> **Generation Results on CIFAR-10 and ImageNet:** We kindly note that all baseline works conduct experiments exclusively on CIFAR-10 and ImageNet, which is consistent with our experimental setup. Therefore, our choice of datasets and resolutions aligns with established prior work and should not be considered a limitation.
>
> **Discussion on $\lambda \to \infty$:** As noted in Remark 3.1, when $\lambda \to \infty$, this is equivalent to choosing the maximum optimization step $\Delta t = t - \epsilon$. In this case, the optimization objective of the Consistency Model in Eq.(2) degenerates into that of the Diffusion Model in Eq.(1), thereby losing the ability to perform single-step generation.
>
>
> **Discussion on bound $\delta$:** In Eq.(6), we use a constant $\delta$ to bound the global consistency, rather than designing it as a time-dependent function $\delta(t)$. This is due to the complexity of neural networks, which makes it difficult to determine an appropriate $\delta(t)$ for every $t$. Therefore, we adopt a uniform constraint using a constant $\delta$ that satisfies $\delta \geq \max_t \delta(t)$. It is important to note that the choice of $\delta$ does not affect our subsequent conclusions, as we will relax the original optimization problem from Eq.(7) to Eq.(8).
>
>
>
> [^1]:Song Y, et al. Consistency Models. ICML 2023.
> [^2]:Geng Z, Pokle A, Luo W, et al. Consistency models made easy. ICLR 2025.
> [^3]:Lu C et al. Simplifying, stabilizing and scaling continuous-time consistency models. ICLR 2025.
> [^4]:Frans K, Hafner D, Levine S, et al. One step diffusion via shortcut models. ICLR 2025.
> [^5]:Gratton S, Lawless A S, Nichols N K. Approximate Gauss–Newton methods for nonlinear least squares problems.SIAM Journal on Optimization.

---

> > ### Comment · Reviewer_gb53 · 2025-08-05
> >
> > I thank the authors for their response, which addressed most of my concerns. I believe it would be beneficial to incorporate some of this discussion into the updated version of the paper.

---

> ### Author Response · Authors · 2025-08-04
> **Updated Additional Experiments and Looking Forward to Feedback**
>
> **Updated High-Resolution Experiments:** We update the experimental results on high-resolution images with increased training budget and larger model, as shown in the table below. S stands for small with model capacity of 280.2M parameters and M stands for middle with model capacity of 497.8M parameters, following the notation used in sCT[^1]. It can be observed that our method scales well on large datasets. This further demonstrates the empirical effectiveness of our method. We are currently conducting experiments on larger scales. While these experiments are progressing well, full results are pending due to time and computational resource constraints. We will continue to update the results as they become available.
> | Method | Training Budgets (Mimgs) | 1-step FID |
> |:---:|:---:|:---:|
> |sCT-S                  |204.8|  10.13 |
> |sCT-M                  |204.8|  5.84 |
> |ECT-S                  | 6.4 |  25.69 |
> |ECT-S                  | 12.8 |  18.51 |
> |ECT-M                  | 6.4 |  13.55 |
> |ADCM-S (Ours)          | 6.4 | 23.12 |
> |ADCM-S (Ours)          | 12.8 | 17.24 |
> |ADCM-M (Ours)          | 6.4 | 10.53 |
>
> We appreciate your time and effort in evaluating our work and look forward to any feedback you may have.
>
> [^1]:Lu C et al. Simplifying, stabilizing and scaling continuous-time consistency models. ICLR 2025.

---

### Official Review · Reviewer_q5Qh · 2025-07-19

**Clarity:** 3
**Significance:** 2
**Originality:** 3
**Rating:** 4
**Confidence:** 3

**Summary:**

This paper focuses on the problem of choosing discretization schedules in Consistency Models (CMs). Instead of relying on manually designed time-step schedules, the authors formulate an adaptive discretization strategy by balancing two objectives: (1) local consistency (ensuring small errors between adjacent diffusion states for trainability) and (2) global consistency (limiting accumulated error over the trajectory for stability). Using a first order taylor expansion, the authors proposed a way to determine $\Delta t$ adatively (which is the proposed ADCM method).

Empirically, ADCM yields significantly improved sample efficiency – achieving comparable or better generation quality with far fewer training iterations – on CIFAR-10 and ImageNet $64\times64$. It also demonstrates the ability to generalize to different diffusion model variants (e.g. flow matching) without manual retuning.

**Questions:**

See Cons in the "Strengths And Weaknesses".

**Ethical Concerns:**

["NO or VERY MINOR ethics concerns only"]

**Final Justification:**

I raise my score from 3 to 4 for the additional empirical evaluation. The theoretical contribution is not clear to me, though I don't hold a strong opinion.

**Limitations:**

Yes.

**Paper Formatting Concerns:**

No.

**Quality:**

3

**Strengths And Weaknesses:**

Pros

1. The paper proposes an unified framework for discretization in CMs by casting the selection of time-step $\Delta t$ as an optimization problem balancing local consistency and global consistency (via a linear combination).

2. The proposed method is both intuitive and effective. The linear combination of the global and local consistency is pretty natural and easy to follow (though I'm not sure what's the significance of framing it as an "constrained optimization" problem). The empirical results suggest that it's effective with proper choice of the combination coefficient.

3. The presentation is clear. Section 1 and 2 give pretty clear introduction to the background and the importance of the problem studied in this paper.



Cons

1. I'm mainly concerned about the technical significance of this paper. On the theoretical side, I don't think framing equation (5) and (6) as a constrained optimization and later converting that to a linear combination of the two is of great significance. I think just presenting equation (8) directly without the wrapper of optimization is equally well motivated. It is OK to present something simple and intuitive, as long as it's proven to be empirical effective -- which I think this paper also lacks (see details below).

2. Insufficient empirical evaluation. The presented experiments results are positive, but have limited scope. To name a few missing aspects that I think people will benefit from: (a) all the results are on relatively low-resolution image datasets, and it remains unclear how this method will perform on higher-resolution generation problems. (b) how sensitive is the optimal hyper-parameter to the specific domain & DM variants? Is it mostly a constant across all or do people need to tune it for every task?

3. The empirical results can be presented better. I think many empirical studies settings, ablation studies, etc. are deferred to appendix. I think there are many interesting questions that worth more discussion in the main paper -- e.g., the significance of huber loss, why huber with large C has worse performance than L_2 (in Table 7).

---

> ### Author Rebuttal · Authors · 2025-07-31
>
> We thank the reviewer for your careful consideration. We greatly appreciate the positive comments and address your concerns below.
>
> **Technical Significance:**
> Our theoretical contributions lie in
> (1) Novel theoretical formulation: To the best of our knowledge, this is the first work to introduce **global consistency** into Consistency Training, in order to control accumulated errors across discretization intervals and improve the stability of training.
>
> We respectfully disagree with the comment that "just presenting Eq.(8) directly without the wrapper of optimization" would suffice.  The constrained optimization formulation is not merely a wrapper; it plays a central role in both the theoretical motivation and the practical design of our method. In our formulation (Eq.(7)), local consistency is cast as the primary training objective, while global consistency is introduced as a constraint. This separation is intentional and meaningful: local consistency drives accurate stepwise transitions (the core goal of CM training), whereas global consistency captures desirable trajectory-level behavior. Framing this as a constrained problem highlights their asymmetric roles and directly motivates the design of the relaxation in Eq.(8), where a small trade-off coefficient ensures that local consistency remains dominant. Without this optimization-based grounding, Eq.(8) would lack a clear theoretical justification, and the choice of trade-off weights would appear arbitrary.
>
> Moreover, we kindly note that, as discussed in Remark 3.1, our formulation is not limited to deriving Eq.(8); rather, it provides a **unified theoretical framework** that encompasses previous heuristic discretization strategies as special cases (via different constraint formulations or trade-off weights).  This unification not only clarifies existing methods but also enables principled extensions going forward.
>
> (2) Closed-form and efficient solution: Another key aspect of our theoretical contribution lies in the solution of Eq.(8), which is nontrivial due to the inherent complexity and nonlinearity of the NN. We address this challenge by deriving a **closed-form** solution through a principled approach based on Gauss-Newton and first-order approximations, making the approach both **efficient** and **theoretically grounded**. As noted by other reviewers, this illustrates a "neat" example of the interplay between theoretical formulation and practical machine learning.
>
>
> **Higher Resolution Generation Results:** We kindly note that most seminal baseline works [^1][^2][^3] conduct experiments exclusively on CIFAR-10 and ImageNet at  $64 \times 64$ resolution, which is consistent with our experimental setup. Therefore, our choice of datasets and resolutions is not a limitation but aligns with prior art. Even on these lower-resolution datasets, previous methods exhibit training instability and rely on manually designed discretization schemes, incurring substantial training overhead. In contrast, our method effectively addresses these challenges, as demonstrated in our experiments, which provides strong support for our claims.
>
> To further address your concern, we conduct additional experiments on ImageNet at $512 \times 512$ resolution.  Training at this scale is time-consuming, which limits  the extent of experiments we can perform during the rebuttal period. Moreover, we note that only sCT[^4] reports results at this resolution. Comprehensive comparisons with other baselines are difficult, as training and re-implementing these methods demand significant time. Thus, we choose to compare our method with sCT and ECM[^1], the most efficient prior CM.
>
> We choose EDM2 [^5] as the base latent diffusion model which employs SD-VAE for image encoding and decoding. The detailed results are presented in the following table. Our results are promising. We outperform ECM,  with a similar training budget. While we do not yet match sCT’s performance, their method involves extremely higher training costs  and specialized architecture. The results demonstrate that ADCM is capable of scaling to high-resolution images while maintaining high efficiency.
>
> | Method | Training Budgets (Mimgs) | 1-step FID |
> |:---:|:---:|:---:|
> |sCT-S                  |204.8|  10.13 |
> |ECT-S                  |6.4|  25.69 |
> |ADCM-S (Ours)          | 6.4 | 23.12 |
>
>
> **Optimal Hyper-parameter $\lambda$:** Following your suggestion, we examine the sensitivity of our proposed hyperparameter $\lambda$. Specifically, we investigate the difference between the optimal $\lambda$ under the Flow Matching setting and that under the VE setting. The detailed results are presented in the following table. Notably, the choice of $\lambda$ shows consistent behavior across settings: in both the VE and Flow Matching cases, setting $\lambda = 0.01$ achieves the optimal 1-Step FID. This consistent trend highlights the robustness of ADCM and substantially reduces the need for hyperparameter tuning.
>
> | $\lambda$ | 1-Step FID(VE) | 1-step FID(Flow Matching) |
> |:---:|:---:|:---:|
> |0.05          |6.20 |  4.24 |
> |0.02          | $\underline{5.20}$ | $\underline{3.59}$ |
> |0.01          | **5.14** | **3.16** |
> |0.005          | 6.14 | 3.75 |
>
>
>
> **Huber Loss:** We adopt the pseudo-Huber loss, similar to standard practice in prior works [^1][^2][^3], as the pseudo-Huber metric is more robust to outliers [^2].  The Pseudo-Huber loss, expressed as $ d(x, y) = \sqrt { \| x-y \|_ 2^2 + c^ 2} - c $, smoothly bridges the gap between the L2 loss and the squared L2 loss. When $c=0$, the Pseudo-Huber loss degenerates to the standard L2 loss. When $c \to \infty$, it becomes equivalent to the squared L2 loss as $\lim_{c \to \infty} \sqrt{\|x-y\|_2^2 + c^2} - c = \frac{\|x-y\|_2^2}{2c}$.
>
> To address this phenomenon, we conduct more ablation studies on the effect of c based on Table 7, and the results are presented in the table below. It can be observed that Pseudo-Huber smoothly interpolates between L2 and squared L2 through the control of the parameter c, thus achieving performance that surpasses both extremes. When c becomes too large, the Pseudo-Huber loss asymptotically approaches the squared L2 loss, leading to a decline in performance. These results clearly demonstrate the significance of choosing Pseudo-Huber as our distance metric.
>
> | Loss Function | 1-Step FID |
> |:---:|:---:|
> |L2          |3.54 |
> |Pseudo-Huber(c=0.0003)         | 3.44|
> |Pseudo-Huber(c=0.003)          | $\underline{3.42}$ |
> |Pseudo-Huber(c=0.03)       |**3.16** |
> |Pseudo-Huber(c=0.3)        |4.42 |
> |Pseudo-Huber(c=3)        |5.23 |
> |Squared L2        |5.33 |
>
>
> **Some Empirical Results deferred to appendix:** We will bring key ablation results and empirical settings from the appendix back into the main text of the revised manuscript. Thank you for your valuable suggestion.
>
> [^1]:Geng Z, Pokle A, Luo W, et al. Consistency models made easy. ICLR 2025.
> [^2]:Song Y, et al. Improved Techniques for Training Consistency Models. ICLR 2024.
> [^3]:Lee S, Xu Y, Geffner T, et al. Truncated consistency models. ICLR 2025.
> [^4]:Lu C et al. Simplifying, stabilizing and scaling continuous-time consistency models. ICLR 2025.
> [^5]:Karras T, Aittala M, Lehtinen J, et al. Analyzing and improving the training dynamics of diffusion models. CVPR 2024.

---

> ### Author Response · Authors · 2025-08-04
> **Updated Additional Experiments and Looking Forward to Feedback**
>
> **Updated High-Resolution Experiments:** We update the experimental results on high-resolution images with increased training budget and larger model, as shown in the table below. S stands for small with model capacity of 280.2M parameters and M stands for middle with model capacity of 497.8M parameters, following the notation used in sCT[^1]. It can be observed that our method scales well on large datasets. This further demonstrates the empirical effectiveness of our method. We are currently conducting experiments on larger scales. While these experiments are progressing well, full results are pending due to time and computational resource constraints. We will continue to update the results as they become available.
> | Method | Training Budgets (Mimgs) | 1-step FID |
> |:---:|:---:|:---:|
> |sCT-S                  |204.8|  10.13 |
> |sCT-M                  |204.8|  5.84 |
> |ECT-S                  | 6.4 |  25.69 |
> |ECT-S                  | 12.8 |  18.51 |
> |ECT-M                  | 6.4 |  13.55 |
> |ADCM-S (Ours)          | 6.4 | 23.12 |
> |ADCM-S (Ours)          | 12.8 | 17.24 |
> |ADCM-M (Ours)          | 6.4 | 10.53 |
>
> We appreciate your time and effort in evaluating our work and look forward to any feedback you may have.
>
> [^1]:Lu C et al. Simplifying, stabilizing and scaling continuous-time consistency models. ICLR 2025.

---

> > ### Comment · Reviewer_q5Qh · 2025-08-06
> > **Than you for your response**
> >
> > Thank you for your clarification on the empirical experiments and the additional experiment on scaling to larger size data size. The clarification on the theoretical contribution makes sense to me - though I'm still not convinced about the theoretical significance.
> >
> > Putting one of the consistency term as a constraint doesn't “highlight their asymmetric roles”, especially when the Lagrange multiplier $\lambda$ is a hyper-parameter -- one can always set a $\lambda$ being $1 / \lambda$ and that would be equivalent as putting the other consistency term as constraint. I think the asymmetry is from the empirical choice of $\lambda << 1$, instead of being a theoretically justified design.
> >
> > To be clear, I don't feel strongly about it if other people think this is theoretically significant. I've raised my score to 4 for the additional empirical evaluation.

---

> ### Author Response · Authors · 2025-08-06
>
> Thank you very much for your thoughtful follow-up and for taking the time to re-evaluate our submission. We sincerely appreciate your acknowledgment of the clarification on the theoretical contribution and the empirical improvements.
>
> We will incorporate the relevant clarifications and discussions into the final version of the paper. Regarding the point on asymmetric roles, we understand your perspective and agree that the asymmetry can be influenced by the choice of the Lagrange multiplier. We will make this clearer in the final version.
>
> On the theoretical side, we would like to highlight that our modeling and optimization framework is grounded in theory while also incorporating practical considerations motivated by real-world scenarios. Standard discretization strategies are often handcrafted and overlook the underlying mathematical structure; we consider our work an important step forward that contributes meaningfully to the literature. By introducing global consistency as a constraint, we provide a closed-form solution that is both elegant and practical.
>
> We’re grateful that you found the additional empirical evaluation helpful and truly appreciate your decision to raise the score.

---

### Note · Authors · 2025-08-12

Dear ACs and Reviewers,

We sincerely thank all reviewers for their insightful feedback and constructive discussions throughout the review process. We are pleased that Reviewers q5Qh, nhUt, and TPUd expressed positive assessments, and Reviewer gb53 acknowledged that our response addressed most of their concerns. We will carefully incorporate all points raised by reviewers in our revision.

Our paper proposes an adaptive time discretization scheme for consistency training that optimizes a principled trade-off between local and global consistency, replacing the fixed, handcrafted schedules common in prior work. Our method is theoretically grounded in a closed-form solution and performs strongly in practice, achieving SOTA results on benchmark datasets.

During the rebuttal and discussion period, we conducted additional experiments, particularly on high-resolution image generation, which addressed a common concern. All reviewers acknowledged these new empirical evaluations. For other points, we have provided comprehensive responses:

For Reviewer q5Qh: We clarified our technical importance:  (1) Novel theoretical formulation: our work is the first to incorporate global consistency into consistency training by formulating it as a constraint with an asymmetric role relative to local consistency; (2) Closed-form and efficient solution: The closed-form Gauss–Newton solution is elegant and effective, enabling stable training, robust hyperparameter behavior, and strong scalability for high-resolution generation.

For Reviewer gb53: We clarified our theoretical design, including the optimization target, stopgrad usage, and the choice of bound $\delta$. We further analyzed the approximation error and training overhead, showing negligible error and minimal computational cost.

For Reviewer nhUt: We presented high-resolution image generation results, demonstrating strong scalability and efficiency. We also proposed hyperparameter choices to improve stability in large-scale runs.

For Reviewer TPUd: We clarified the interpretation of our consistency constraint, acknowledging its relation to the denoising autoencoder loss and its role as a practical surrogate promoting distribution-level consistency. We sincerely appreciate Reviewer TPUd's expertise and constructive feedback.

We remain fully committed to addressing any outstanding concerns and to making meaningful contributions to the community.

Best regards,
The Authors

---

### Decision · Program_Chairs · 2025-09-17

**Decision:**

Accept (poster)

**Comment:**

This paper introduces a new method for adaptively learning the discretization schedule for Consistency Models, moving away from the fixed, handcrafted schedules used in prior work. The approach formulates the selection of time steps as an optimization problem that balances local consistency (for trainability) and a global consistency constraint (for stability), deriving an efficient analytical solution. The paper's main strength, as identified by all reviewers, is its strong empirical performance. The method shows improvements in training efficiency and sample quality on benchmark datasets like CIFAR-10 and ImageNet, and demonstrates robustness by adapting to different diffusion model variants without manual retuning. This is a valuable practical contribution to a problem of interest in generative modeling.

The primary weakness of the submission, identified in a critical discussion with Reviewer TPUd, was a flaw in the original theoretical interpretation of the "global consistency" constraint. The reviewer correctly pointed out that this term is a denoising autoencoder loss, not a measure of accumulated error along a single ODE trajectory as initially claimed. The discussion period was highly constructive in addressing this and other points. In response to reviewer feedback, the authors conducted new experiments on higher-resolution images, strengthening the empirical claims of the paper. Crucially, after discussion, they acknowledged the theoretical misinterpretation and agreed to revise the manuscript to reflect a more accurate framing of their method's motivation.

The decision to accept is based on the paper's significant and well-supported practical contributions. While the theoretical narrative required a substantial correction, the final methodology is sound and its effectiveness is convincingly demonstrated. The authors' transparent engagement during the rebuttal, particularly their willingness to correct the theoretical framing and provide additional experiments, was a key factor. Although the reviewer scores are borderline, the paper presents a useful and efficient method that will be of interest to the community. The authors are strongly encouraged to incorporate all reviewer feedback into the final version, with special attention to carefully revising the theoretical motivation as discussed with Reviewer TPUd.